# Synergistic elastase and papain injury drives abdominal aortic aneurysm formation and rupture in mice

Santiago Elizondo-Benedetto [1], Mohamed S. Zaghloul[1], Batool Arif[1], Ibrahim Kuziez[1], Ryan Wahidi[1,2] & Mohamed A. Zayed [1,2,3,4,5,6,7] ✉

## Abstract

**Background** Abdominal aortic aneurysm (AAA) rupture leads to high morbidity and mortality. Current rodent models struggle to reliably mimic infrarenal AAA rupture. Chemical treatments using pancreatic elastase (PE), papain (Pa), β-aminopropionitrile (BAPN), and angiotensin II (ANG II) are known to induce AAA in rodents. We hypothesized that combining these agents can synergistically lead to acute AAA rupture models, as well as chronic AAA models that closely resemble human pathology.

**Methods** AAAs were induced in 125 male C57BL/6 mice via peri-adventitial exposure for twenty minutes using a cotton ball with either PE, Pa, or a combination of both (PE+Pa), with or without BAPN and ANG II.

**Results** Two weeks post-induction, all groups exhibit significantly elevated aortic diameters, increased inflammation, elastin and collagen degradation, and matrix metallopeptidase (MMP) activity. The addition of BAPN results in large chronic AAAs (500% growth) and intraluminal thrombus (ILT) formation. Further addition of ANG II results in a 93% rupture rate in the PE+Pa group, significantly increased compared to PE and Pa alone. Compared to previous models, the PE+Pa, BAPN and ANG II combination demonstrates an increase in rupture events, inflammation, and MMP activation.

**Conclusions** This murine model, using a synergistic combination of pancreatic elastase and papain, effectively replicates AAA pathophysiology and is ideal for investigating underlying mechanisms and potential therapeutic interventions.

## Plain Language Summary

Abdominal aortic aneurysm (AAA) is a serious condition where the main blood vessel in the abdomen enlarges dangerously and eventually ruptures, causing high death rates. Current mouse models struggle to mimic this condition accurately. Our study tested a combination of chemicals, including pancreatic elastase, papain, β-aminopropionitrile (BAPN), and angiotensin II (ANG II), to develop a model that better replicates human AAA disease. A novel combination of pancreatic elastase and papain caused significant aortic enlargement, increased inflammation, and tissue degradation similar to what is observed in humans. Adding BAPN results in large chronic AAAs and blood clots, while adding ANG II leads to a high AAA rupture rate. This new model could facilitate the development of improved AAA prevention and treatment in humans.

Abdominal aortic aneurysm (AAA) is a life-threatening vascular condition characterized by the localized dilation of the abdominal aorta, which can progress to rupture leading to fatal events[1,2]. Despite advances in clinical management, AAA rupture continues to be a leading cause of mortality, resulting in an annual mortality rate of approximately 150,000 worldwide[3,4]. The pathophysiology of AAA involves intricate biomolecular processes, with inflammation playing a crucial role in its progression[2,5]. Inflammatory cells, such as macrophages and T-lymphocytes, infiltrate the aortic wall and release proteolytic enzymes, including matrix metalloproteinases (MMPs),

which contribute to the degradation of the extracellular matrix (ECM)[6–9]. This inflammatory response, often triggered by risk factors such as hypertension and smoking, disrupts the balance between proteolytic activity and ECM synthesis, leading to progressive dilation of the aorta[10]. However, the exact mechanisms underlying AAA progression and rupture remain unclear, highlighting the need for reliable pre-clinical models to study disease signaling mechanisms and evaluate potential novel therapies aimed at preventing AAA progression and rupture.

[1]Division of Vascular Surgery, Department of Surgery, Washington University School of Medicine, St. Louis, MO, USA. [2]Cardiovascular Research Innovation in Surgery and Engineering Center, Department of Surgery, Washington University School of Medicine, St. Louis, MO, USA. [3]Department of Radiology, Washington University School of Medicine, St. Louis, MO, USA. [4]Division of Molecular Cell Biology, Washington University School of Medicine, St. Louis, MO, USA. [5]Division of Surgical Sciences, Department of Surgery, Washington University School of Medicine, St. Louis, MO, USA. [6]Department of Biomedical Engineering, McKelvey School of Engineering, Washington University School of Medicine, St. Louis, MO, USA. [7]Veterans Affairs St. Louis Health Care System, St. Louis, MO, USA. ✉e-mail: zayedm@wustl.edu

Genetically-modified murine models have become essential for investigating AAA pathophysiology[11–13]. These models aim to replicate key features of human AAA disease, such as aortic dilation, ECM degradation, intraluminal thrombus (ILT) formation and inflammation. Widely used models include genetic variants like *Apoe*[-/-] mice and chemically induced models. In these traditional mouse models, chemical induction of AAAs can be achieved by using pancreatic elastase (a proteolytic enzyme that degrades elastin)[14], β-aminopropionitrile (BAPN; a non-specific inhibitor of lysyl oxidase that disrupts collagen cross-linking)[15], and angiotensin II (ANG II; a peptide hormone that induces hypertension)[16,17]. Recently it was observed that papain, a cysteine protease derived from papaya latex that also degrades elastin, can also induce AAA formation in rodents[18]. Although these models offer valuable insights into the molecular and cellular mechanisms of AAA disease, they often fail to fully recapitulate the complexity of human AAAs. This includes challenges in accurately representing body-adjusted AAA diameter, chronicity, aortic anatomical location, and rupture incidence. For example, on their own pancreatic elastase and papain only induce localized elastin degradation, and fail to replicate the systemic inflammatory and hemodynamic components that resemble human AAA disease progression[14,18,19]. In contrast, the Ang II model primarily causes aneurysms in the suprarenal aorta, and fails to replicate the infrarenal AAA disease pathology. Moreover, the reliance on hyperlipidemic mice, like *Apoe*[-/-] mice, limits the applicability of these models to normolipidemic AAA patients, which is a significant drawback[11,16].

A major challenge in AAA investigations is the failure to translate preclinical findings into effective clinical therapies. Despite the promising results of numerous pharmacological agents in established mouse AAA models, none have progressed to FDA treatments for AAAs in humans[20]. These translational gaps may be attributed to the lack of standardized protocols for inducing and evaluating AAAs in mice, and inadequate disease modeling of human pathology, which unfortunately causes variability in experimental outcomes[21]. Therefore, establishing a standardized animal model that closely resemble human AAA disease pathology is crucial for understanding the impact of possible future therapies and improve translational success in humans.

In this study we demonstrate that combining specific factors known to induce aneurysm growth exhibit a synergistic effect on AAA development. The novel combination of pancreatic elastase and papain (PE+Pa) significantly influences AAA expansion, histopathology, and inflammation over a 14-day period. Further enhancing this with BAPN administration over 42 days leads to the formation of chronic AAAs and distinct tissue inflammatory signals, underscoring the model's utility for long-term studies. Lastly, the addition of ANG II pump achieves a remarkable 93% rate of infrarenal AAA ruptures. This model replicates key human AAA characteristics, enabling reliable measurements of AAA expansion with minimal technical and procedural complications. It is a valuable tool for both acute and chronic AAA mechanistic investigations.

## Methods
### Animals
Adult 6–8-week-old, male mice on a C57BL/6 background ($n = 125$) were obtained from The Jackson Laboratory (Bar Harbor, ME). All animals were housed at 21 °C in a 12/12 h light/dark cycle and had access to food and water ad libitum. Anesthesia was administered with a mixture of ~1.5% isoflurane and oxygen for all procedures (Supplementary Table 1). The core body temperature was monitored and maintained with a heating pad (37 °C). Use of all animal experiments were performed in accordance with relevant guidelines and regulations and approved by the Institutional Animal Care and Use Committee (IACUC) at Washington University School of Medicine in St. Louis. At the conclusion of studies, live animals were euthanized following IACUC protocols.

### Induction of AAA and normal saline-control models
Male mice were induced to develop infrarenal AAAs using either pancreatic elastase[14](PE; 10.3 mg protein/mL, 5.9 U/mg protein obtained from Sigma

Aldrich), papain[18] (Pa; 1.0 or 20 mg/mL) or a combination of PE and Pa (PE +Pa). Surgical positive controls were exposed to normal saline (NS), while negative or wildtype (WT) surgical controls did not undergo any surgical procedure (Fig. 5). Ventral abdominal wall laparotomy was performed, and the infrarenal abdominal aorta exposed, from the left renal vein to the aortic bifurcation. As previously reported, a sterile cotton ball ($3.0 \times 5.0$ mm) was embedded with 50 μL of a 100% concentration of either PE, Pa or NS using a pipette with a fine tip and placed on top of the aorta to successfully isolate the chemical exposure from the surroundings[18]. For the PE+Pa combination, an additional step is included: Pa is applied first for 10 min, followed by a quick washout period, and then PE is applied for another 10 min. After 20 min of either chemical incubation, the abdominal cavity is washed twice with saline to remove any remnant papain or pancreatic elastase (Supplementary Video 1). The ventral abdomen is then closed in a continuous and interrupted fashion in two layers, with muscle and fascia closed with a 5-0 Vicryl suture, and skin closed with a 5-0 nylon monofilament suture respectively (Supplementary Table 1). Most surgeries were conducted within 30–45 min by two surgeons, blinded to group allocation and analysis, although groups were not randomized. Using the Leica IVESTA 3 video micrometer, the baseline maximum aortic diameter was measured. After 14 days (Fig. 1A and Fig. 5), mice aortas were re-exposed via ventral abdominal laparotomy, maximal aortic diameters were measured, and aortic tissue was harvested for further analysis. Aortic aneurysms were defined as >50% increase in the aortic maximum diameter relative to baseline diameter ($\approx 0.5$ mm).

### Chronic model of AAA in male C57BL/6 mice
In addition to either PE, Pa or PE+Pa chemical incubation, starting 3 days before chemical exposure and daily thereafter, mice also underwent β-aminopropionitrile (BAPN) administration through drinking water (0.3% BAPN in water) to promote AAA rupture (Fig. 2A, Supplementary Fig. 3A and Fig. 5)[15,22]. Mice were ≈25 g and drank 2.5 mL water/day, leading to intake of ≈30 mg BAPN/ (kg·day; Supplementary Fig. 3B, C). After 42 days (6 weeks), all mice aortas were re-exposed via ventral abdominal laparotomy, maximal aortic diameters were measured, and aortic tissue was harvested for further analysis (Fig. 2A, Supplementary Fig. 3A and Fig. 5).

### AAA rupture induction
In addition to either PE, Pa or PE+Pa chemical incubation and BAPN administration, a group of 52 mice underwent subcutaneous osmotic pump implantation (Alzet 1004, Durect Corp, Cupertino, CA) to elute angiotensin II (ANG II; Sigma Aldrich Inc, St. Louis, MO) at 2000 ng/kg/min for 14 days as previously described[17,22], to assess the synergistic effects of these combinations on infrarenal AAA rupture (Fig. 3A, Fig. 5 and Table 1). Mice that developed ruptured AAAs during the study period promptly underwent necropsy to confirm and analyze the pathology (Fig. 3D), while those that did not rupture by day 14 were identified as the non-ruptured AAAs and humanely sacrificed for tissue analysis (Fig. 3E). For comparative analysis, outcomes from the novel PE+Pa combination were evaluated relative to a previously described AAA model[19]. In this model, BAPN administration began 3 days prior to AAA induction. The procedure involved the topical application of 5 μL 100% concentration of PPE for 5 minutes (5'PE, $n = 15$), without using a cotton ball. After either 6 or 14-days' time points, mice were sacrificed, AAA diameters were evaluated, and aortic tissue was harvested for further analysis (Fig. 4, Fig. 5, and Table 1).

### Postoperative analgesia & euthanasia
Mice were treated with pre-operative analgesia (Buprenorphine SR) 1 hour prior to the AAA induction procedure. Buprenorphine SR provides 72 hours of pain relief and was dosed at 1 mg/kg and administered subcutaneously. At the time of euthanasia, animals were anesthetized with isoflurane, and the ventral abdominal incision was reopened. The AAA was dissected free from the surrounding tissue, and blood was collected from the inferior vena cava using an insulin syringe ($0.33 \times 12.7$ mm) as summarized in Supplementary Table 1. The aorta was then excised from the level of the

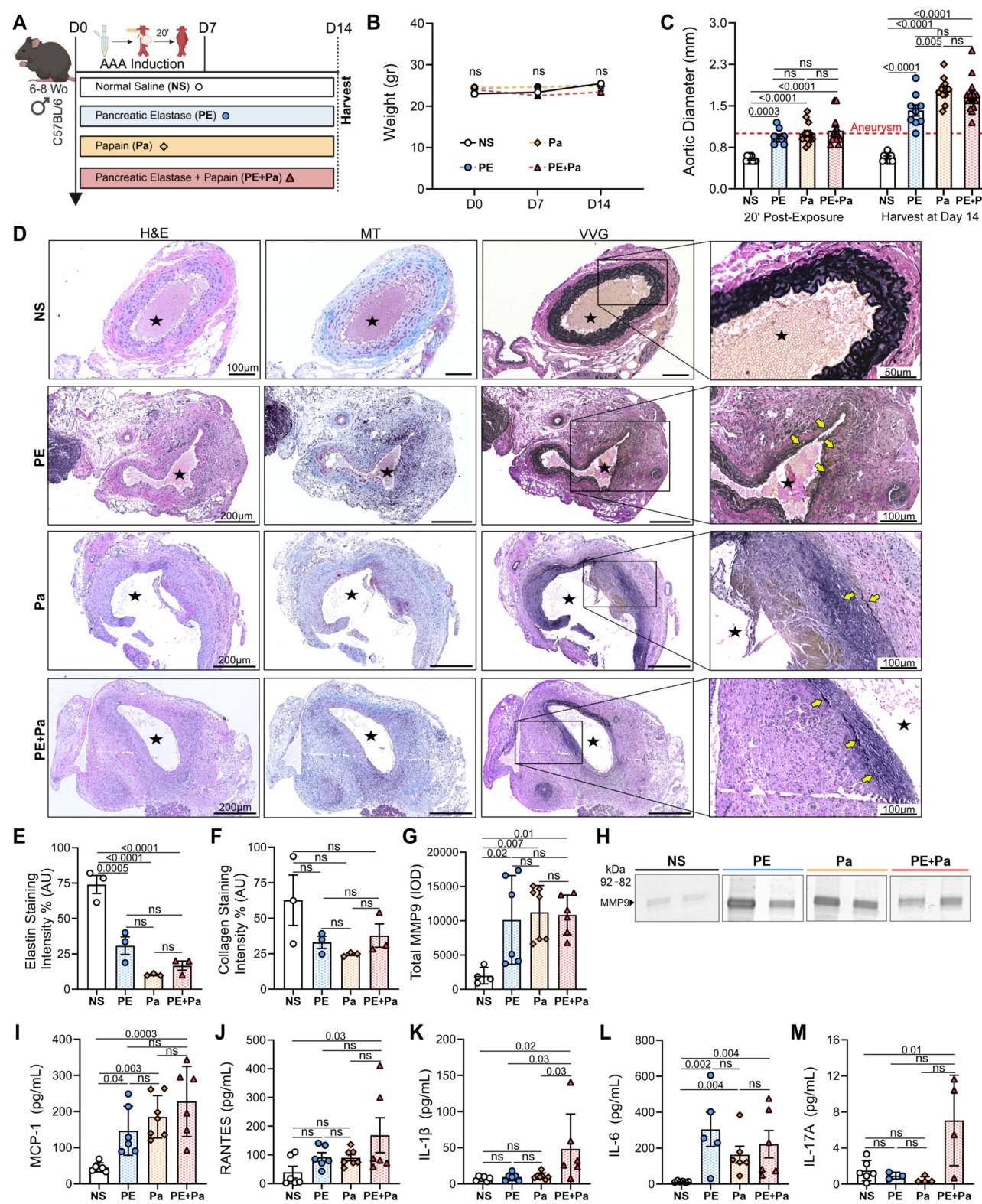

left renal vein to the aortic bifurcation for tissue processing and analysis. The thoracic aorta was also collected and processed accordingly for the studied groups.

## Animal weight and BAPN consumption
Mice whole body weights were evaluated either at day 0 pre-AAA induction, or from week 1–6 followed AAA induction. Absolute numbers in grams were evaluated and differentiated among studied groups to assess tolerability to the chemical exposure and procedure as well as recovery process. Not significant changes were noted.

## Histology and immunostaining of mice AAA tissue sections
Aortic tissue was harvested from all animals. AAA tissue was fixed in HistoChoice (VWR), and paraffin embedded. Paraffin blocks were

**Fig. 1 | Day 14 AAA formation following peri-adventitial aortic exposure to either PE, Pa or the PE + Pa combination.** **A** Mice underwent exposure to either normal saline (NS, *n* = 10), pancreatic elastase (PE, *n* = 9), papain (Pa, *n* = 13) or a combination (PE+Pa, *n* = 14) to assess AAA formation. All mice aortas were harvested on day 14 for histological or protein assessment from distinct samples. Panel A was made using BioRender.com. **B** Body weight assessment in grams at days 0, 7 and 14 post-AAA development. **C** Aortic diameter evaluation in mm, at day 0 (20 min post-exposure) and at day 14 (post-AAA development). Aneurysms were defined by a size greater than 1 mm (50% increase from baseline measurements). Aneurysms were defined by a size greater than 1 mm (50% increase from baseline measurements). Three (*n* = 3) mice of each group were harvested and processed for histopathological analysis (**D**) H&E, MT and VVG staining of abdominal aortas (cross-section of tissue slides) with 5x and 10x magnification. **E** Quantification of AAA elastin and (**F**) collagen fibers via staining intensity (arbitrary units, AU). A

total of six (*n* = 6) mice from the NS, PE, and PE+Pa groups, and seven (*n* = 7) mice from the Pa group are processed for protein analysis. Two tissue samples from NS were excluded due insufficient space on the zymogram gel. **G** Quantification of total MMP-9 levels via integrated density (IOD). **H** Zymogram demonstrating total MMP-9 activity band. Chemokines (**I**) MCP-1 and (**J**) RANTES and pro-inflammatory markers (**K**) IL-1β, (**L**) IL-6 and (**M**) IL-17A content within the AAA tissue measured by ELISA. 1-2 tissue samples from PE, Pa and PE+Pa were excluded from IL-6 and IL-17A due to insufficient or undetectable readings. Star indicates the aortic lumen. Yellow arrows indicate the elastin fibers. Data are presented as mean ± standard deviation (SD). Ns>0.05 and statistical analysis was performed using one-way ANOVA for single variable, two-way ANOVA with multiple comparison for continuous variables (weight and AAA size) to compare NS vs PE, Pa, and PE+Pa, or the two-tailed Mann-Whitney test when comparing PE, Pa, and PE+Pa individually.

sectioned at 5 μm, and deparaffinized. To evaluate AAA tissue morphology and pathology, tissue sections were evaluated using Hematoxylin and Eosin (H&E), Verhoeff-Van Gieson (VVG), and Mason Trichrome (MT) staining using NanoZoomer HT 2.0 (Hamamatsu Photonics; Shizuoka, Japan), supported by the Alafi Neuroimaging Laboratory (Washington University School of Medicine). MT and VVG cross-section staining were quantitatively analyzed using ImageJ software via color deconvolution, adjusted threshold, and region of interest assessment of the AAA wall to measure collagen and elastin degradation, as previously reported[22]. Additionally, H&E and VVG cross-sections were analyzed in a semi-quantitative manner to assess inflammation, elastin degradation and vascular smooth muscle cell (VSMC) loss within the aneurysm wall, as previously described[23]. Briefly, a histopathological grading system was defined by a clinical pathologist, with three measured parameters: Inflammation (mild = no or minimal cell infiltration; moderate = scatter cells without clustering; severe = clusters of inflammatory cells), Elastin degradation (mild = mostly intact elastin fibers with less than 10% of disruption; moderate = shredded elastin layers with 10–70% of disruption; severe = more than 70% of elastin fiber disruption) and VSMC loss (mild = decreased nucleus density; moderate = 10-50% of decrease in nucleus density; and severe >50% decrease in nucleus density)[23].

### ELISA and cytokine array assessments
Proteins were extracted from AAA tissue using RIPA buffer supplemented with a protease inhibitor (Sigma #MCL1). Protein concentrations were determined via the Bradford assay. For each AAA tissue sample, 25 μg of protein was analyzed. Specific assays conducted included a cytokine multiplex assay (Millipore, RECYTMAG-65K), all according to the manufacturers' instructions.

### MMP2 and MMP9 zymography
For each AAA tissue sample, 25 μg of protein was loaded onto the wells of 10% Gelatin Zymogram electrophoresis gels. The gels were incubated in Zymogram renaturing buffer for 30 min, followed by 36 hours in Zymogram developing buffer at 37 °C. Subsequently, the gels were stained with Coomassie Brilliant Blue R-25 solution (BioRad) for 30 min and then destained using a buffer containing 20% methanol, 20% acetic acid, and 60% deionized water until MMP bands became visible. The gels were scanned using a BioRad ChemiDoc system and analyzed with ImageJ software.

### Statistics and reproducibility
Continuous variables are presented as the mean ± standard deviation (SD), and categorical data as counts and proportions. For comparisons between two groups, a two-tailed Mann-Whitney test was performed. For multiple comparisons that included one endpoint in more than one animal/chemically exposed groups, an ordinary one-way analysis of variance (ANOVA) was applied. For multiple comparisons that included more than one endpoint in more than one animal/chemically exposed group, a two-way ANOVA was performed. Dunnett's multiple comparison test was applied as a post hoc analysis for all ANOVA results. Data was considered statistically significant with $p \leq 0.05$. Kaplan–Meier curve was generated to assess the

survival of BAPN-exposed animals. GraphPad Prism (v.10.6.1, San Diego, CA, USA) was used for all statistical analyses and graphical data representations. The sample size (n) for each experiment is provided in corresponding figure legends.

## Results
### Infrarenal aortic exposure to topical PE, Pa, and its combination (PE + Pa) promotes AAA growth and enhances inflammatory signals
AAA development was initially evaluated over a 14-day period (2 weeks) in four different groups of C57BL/6 mice subjected to a 20-minute peri-adventitial (topical) exposure of the infrarenal abdominal aorta to either normal saline (NS), PE, Pa, or a combination of PE + Pa, as illustrated in Fig. 1A and summarized in Table 1. Mice that receive topical exposure to each chemical exhibit comparable weight gain (Fig. 1B), indicating no significant surgical complications. Compared to the NS group, aortic diameter dilates significantly immediately after 20 minutes of chemical exposure to either PE, Pa or PE+Pa (0.6 ± 0.1 mm vs 0.9 ± 0.1 mm, 0.9 ± 0.1 mm or 1.0 ± 0.1 mm, $p = 0.0003$, $p < 0.0001$ and $p < 0.0001$ respectively; Fig. 1C and Supplementary Fig. 1). Furthermore, aortic diameter continues to grow over a 14-day follow-up in the PE, Pa and PE+Pa groups (1.4 ± 0.3 mm, 1.8 ± 0.2 mm and 1.6 ± 0.2 mm respectively) compared to NS group (0.6 ± 0.1 mm, $p < 0.0001$; Fig. 1C and Supplementary Fig. 1). Notably, the Pa group exhibits the greatest increase in AAA diameter at week 2, which was significantly higher than in the PE group ($p = 0.002$; Fig. 1C) but only slightly higher compared to PE+Pa (p=ns).

Histopathological characterization was performed using hematoxylin and eosin (H&E), Verhoeff-Van Gieson (VVG), and Masson trichrome (MT) stains, as previously described[23]. All chemically induced groups show classic AAA dilation with moderate to severe inflammation and elastin degradation, particularly in the Pa and PE+Pa groups compared to the NS group (Fig. 1D and Supplementary Fig. 2A–C). However, VSMC loss is only mild to moderate in this cohort of mice (Supplementary Fig. 2D).

Quantification of VVG and MT staining, conducted as previously described[22], demonstrates significant elastin fiber breakdown in the aortic wall of the PE, Pa and PE+Pa groups compared to NS group ($p = 0.0005$, $p < 0.0001$ and $p < 0.0001$ respectively; Fig. 1D, E). Additionally, a slight reduction in collagen fibers is observed in the experimental groups compared to the NS group (p=ns; Fig. 1D, F).

Gelatin zymography reveals a significant increase in total MMP9, known to promote AAA formation and rupture[24], in the PE, Pa and PE+Pa groups compared to the NS group ($p = 0.02$, $p = 0.007$ and $p = 0.01$ respectively; Fig. 1G, H and Supplementary Fig. 5). Moreover, the pro-inflammatory chemokine monocyte-chemoattractant protein 1 (MCP-1), typically linked to AAA inflammation, was significantly elevated in the aortic tissue of the PE, Pa and PE+Pa groups compared to the NS group (p = 0.04, 0.003 and 0.0003 respectively; Fig. 1I). Interestingly, RANTES levels are significantly increased only in the PE+Pa group compared to NS group ($p = 0.03$; Fig. 1J). Crucial pro-inflammatory cytokines such as IL-1β, IL-6 and IL-17A are elevated in the aortic tissue

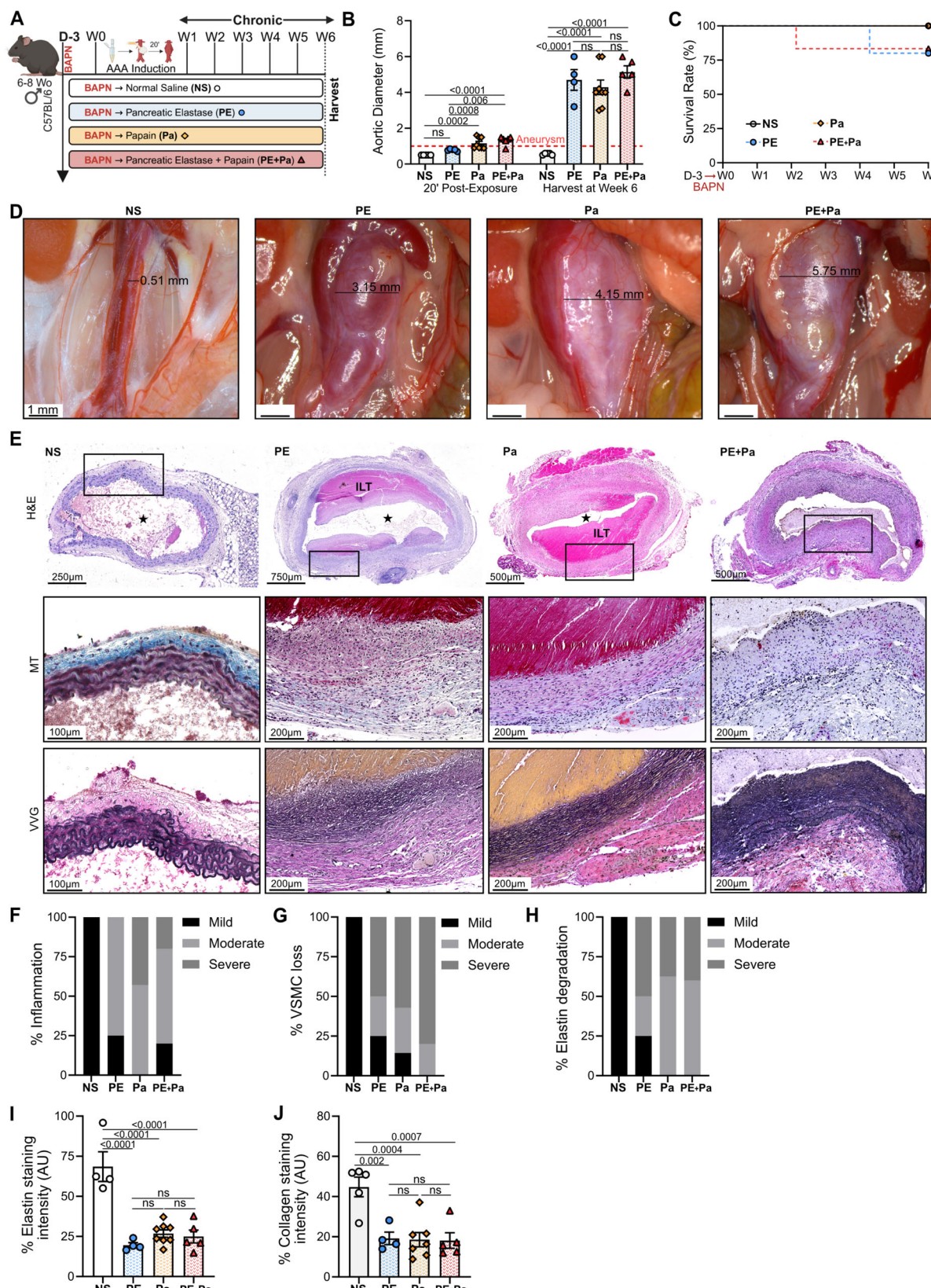

of the PE+Pa group compared to the NS ($p = 0.02$, $p = 0.004$ and $p = 0.01$ respectively; Fig. 1K–M). Moreover, PE+Pa group demonstrated significantly elevated IL-1β levels compared to both PE and Pa groups ($p = 0.03$ and $p = 0.03$, respectively). While the PE and Pa groups also show significantly increased levels of IL-6 ($p = 0.002$ and $0.004$

respectively; Fig.1L), they did not exhibit significant increases in IL-1β and IL-17A compared to NS controls (Figs. 1K, M). Additional cytokines, such as IL-10 and TNF-α, show a slight increase in the PE+Pa group (p=ns), whereas IFN-γ levels are slightly decreased in all three groups compared to the NS group (p=ns; Supplementary Fig. 2E–G).

**Fig. 2 | Chronic model of AAA progression using daily BAPN administration over 6 weeks. A** Mice received BAPN through drinking water starting 3 days prior to AAA induction and continuing until week 6. Mice were also exposed to either normal saline (NS, n = 5), pancreatic elastase (PE, n = 5), papain (Pa, n = 8) or a combination (PE+Pa, n = 6) to promote AAA development. All mice aortas were harvested on day 14 for histological assessment. Panel A was made using BioRender.com. **B** Aortic diameter evaluation in millimeters at day 0 (20 minutes post-exposure) and at day 42 (post-AAA development). Aneurysms were defined by a size greater than 1 mm (50% increase from baseline measurements). **C** Kaplan-Meier curve demonstrating rate of survival following AAA induction to demonstrate infrarenal AAA rupture events. **D** Variable impact of chemically induced AAA development on aortic diameter 6 weeks post-AAA development. **E** H&E, Masson trichrome (MT) and VVG staining of abdominal aortas (cross-section of tissue slides) with 5x and 10x magnification. The star represents the lumen of the artery. ILT = Intraluminal thrombus. IHC categorical analysis (**F**) Degree of inflammation, (**G**) VSMC loss in percent and (**H**) elastin degradation. **I** Quantification of AAA elastin and (**J**) collagen fibers via staining intensity (arbitrary units, AU). One tissue sample from NS was excluded from elastin quantification, and one tissue sample from Pa was excluded from collagen quantification, both due to insufficient quality for accurate measurement. Data are presented as mean ± standard deviation (SD). Ns>0.05 and statistical analysis was performed using one-way ANOVA for single variable (elastin and collagen), two-way ANOVA with multiple comparison for continuous variables (AAA size) to compare NS vs PE, Pa and PE+Pa, or the two-tailed Mann-Whitney test when comparing PE, Pa and PE+Pa individually, and Log rank test for survival rate.

## β-aminopropionitrile enhances impact of topical PE, Pa and PE + Pa for the development of chronic infrarenal AAAs

In another murine cohort, aneurysm progression was evaluated over a 42-day period (6 weeks) following the addition of β-aminopropionitrile (BAPN)[15,22], a well-established ECM repair disruptor extensively used in murine models of AAA formation. As described above, mice underwent a 20-minute topical exposure to either NS, Pa, PE or PE+Pa combination. To further induce AAA progression, 0.3% BAPN was administered daily in drinking water starting 3 days prior to AAA induction (Fig. 2A) and in wild type (WT) mice that received only BAPN without operative AAA induction (Supplementary Fig. 3A). None of the mouse groups demonstrate differences in weight gain or BAPN consumption throughout the experimental timeline (Supplementary Fig. 3B&C).

Compared to the NS group, aortic diameter dilates significantly immediately after 20 minutes of chemical incubation with Pa and PE+Pa (0.5 ± 0.1 mm vs 1.1 ± 0.1 mm and 1.1 ± 0.1 mm, p = 0.0002 and p < 0.0001 respectively) but not with PE (vs 0.9 ± 0.1 mm, p=ns; Fig. 2B). Notably, after 6 weeks of BAPN, the aortic diameter shows consistent growth in the PE, Pa, and PE+Pa groups, significantly exceeding that of the NS group (0.6 ± 0.02 mm vs 4.7 ± 1.2 mm; 5 ± 1.4 mm and 5.1 ± 0.7 mm, respectively; p < 0.0001), representing a fivefold increase over the baseline size of the mouse aorta (Fig. 2B&D). Mice exposed to PE exhibit a 20% rupture rate, while the PE+Pa group demonstrates a 17% rupture rate, and no instances of AAA rupture in the Pa and NS groups (Fig. 2C).

Histopathological characterization reveals typical features of aneurysmal degeneration, with increased inflammation and elastin degradation, similar to the changes observed in the 2-week cohort (Fig. 2E). Conversely, VSMC loss is substantially more severe in this chronic cohort, when compared to the NS group (Fig. 2E–G). Quantification of VVG-staining demonstrates significant elastin breakdown in the aortic wall of all tested groups compared to the NS group (p < 0.0001 Fig. 2E, I). Additionally, quantification of MT-staining indicates a significant decrease in collagen fibers in the aortic walls of PE, Pa and PE+Pa groups, when compared to NS (p = 0.002, p = 0.0004 and p = 0.0007 respectively; Fig. 2E–J). Remarkably, almost half of harvested aneurysms display ILT formation, closely mirroring human AAA disease pathology (Fig. 2E and Supplementary Fig. 4). Recent investigations have shown the development of aneurysmal dissection in the ascending and descending aortas using BAPN in younger mice[25]. Therefore, we evaluated both the thoracic and abdominal aortas of WT mice, after 6 weeks of BAPN exposure and performed histological assessment. We confirm the absence of off-target development of dissections or thoracic aneurysm in this model (Supplementary Fig. 3D).

## Angiotensin II amplifies acute infrarenal AAA rupture, particularly with PE + Pa combined administration

Infrarenal AAA rupture was further evaluated following BAPN administration, this time with the addition of ANG II subcutaneous pumps to induce aneurysm expansion and aortic instability[17,22,26]. Once again, mice underwent a 20 min topical exposure to either Pa, PE or the PE+Pa combination. To evaluate rupture kinetics, a previously described model was replicated using 5 μL of PE for a 5 min topical exposure (5'PE) followed by BAPN administration (Fig. 3A and Table 1)[19]. Mice subjected to the PE+Pa combination demonstrate the highest rupture rate of 93% (13 out of 14 mice), with an average rupture time of 7.4 ± 1.5 days post-AAA induction (Fig. 3B). This is significantly higher when compared to the 5'PE model (p = 0.003), which demonstrate 0% ruptures. Furthermore, the PE+Pa combination also leads to a significant increase in rupture events, with an absolute risk augmentation of 73% and 33% when compared to Pa and PE alone (p = 0.009 and p = 0.03 respectively), despite exhibiting similar post-exposure aortic diameter (Fig. 3C). Additionally, aortic diameter immediately post-exposure is significantly lower in 5'PE group when compared to PE+Pa, PE, and Pa (0.5 ± 0.05 mm vs 0.8 ± 0.07 mm; 0.9 ± 0.06 mm and 0.9 ± 0.1 mm, p = 0.002, p < 0.0001 and p < 0.0001 respectively; Fig. 3C). However, the rupture events for PE and Pa are not statistically significant when compared to 5'PE. Consistent with previous murine models of AAA rupture, post-mortem evaluation of the PE+Pa group consistently reveals retroperitoneal hematomas, a pathognomonic finding of aneurysm rupture events (Fig. 3D). An analysis of the single mouse that survived the PE+Pa combination up to day 14 shows marked increases of abdominal aortic wall cellular infiltration, elastin breakdown, and ILT formation, when compared to the thoracic aorta from the same mouse (Fig. 3E).

## PE + Pa novel combination, with the addition of BAPN and ANG II, reveals increased inflammatory features

To investigate the potential pathophysiological factors associated with the high acute rupture rates observed with the PE+Pa combination, aortic tissue was harvested on day 6 prior to acute AAA rupture events (Fig. 4A). Comparisons were made relative to the 5'PE mouse model, which was established as previously described[19]. The data presented in Fig. 3C were extended with the new cohort to assess aortic dilation between 5'PE and PE+Pa. Compared to the 5'PE group, aortic diameter dilates significantly immediately after 20 minutes of chemical incubation with the PE+Pa combination (0.5 ± 0.05 mm vs 0.9 ± 0.1 mm, p < 0.001) as well as at day 6 (1.0 ± 0.4 mm vs 1.3 ± 0.2 mm, p = 0.0004; Fig. 4B).

Gelatin zymography demonstrates that total MMP2 and MMP9 activity, which are highly associated with AAA formation and rupture, are significantly increased in the PE+Pa group compared to 5'PE group (p = 0.008 and p = 0.001 respectively; Fig. 4C–E and Supplementary Fig. 6). Histopathological analysis shows classic features of de novo aortic degeneration, such as wall dilation and media layer expansion with increased inflammatory cell infiltration at day 6. Dissections, intramural thrombus, and ILT formation are not observed (Supplementary Table 2). Importantly, elastin degradation is increased in both 5'PE and PE+Pa groups (p=ns; Fig. 4F, H) and qualitative analysis demonstrates a more severe degradation in the PE+Pa group (Fig. 4G, H).

Moreover, IL-6 is significantly increased in the PE+Pa group compared to the 5'PE group (p = 0.02; Fig. 4I), while other pro-inflammatory cytokines, such as IL-1β and IL-17A show a modest increase in PE+Pa but are not statistically significant (Fig. 4J, K). Anti-inflammatory IL-10 is unchanged (Fig. 4L). MCP-1 chemokine levels are significantly increased in the AAA tissue of the PE+Pa group compared to 5'PE group (p = 0.02; Fig. 4M), whereas RANTES is unchanged (Fig. 4N).

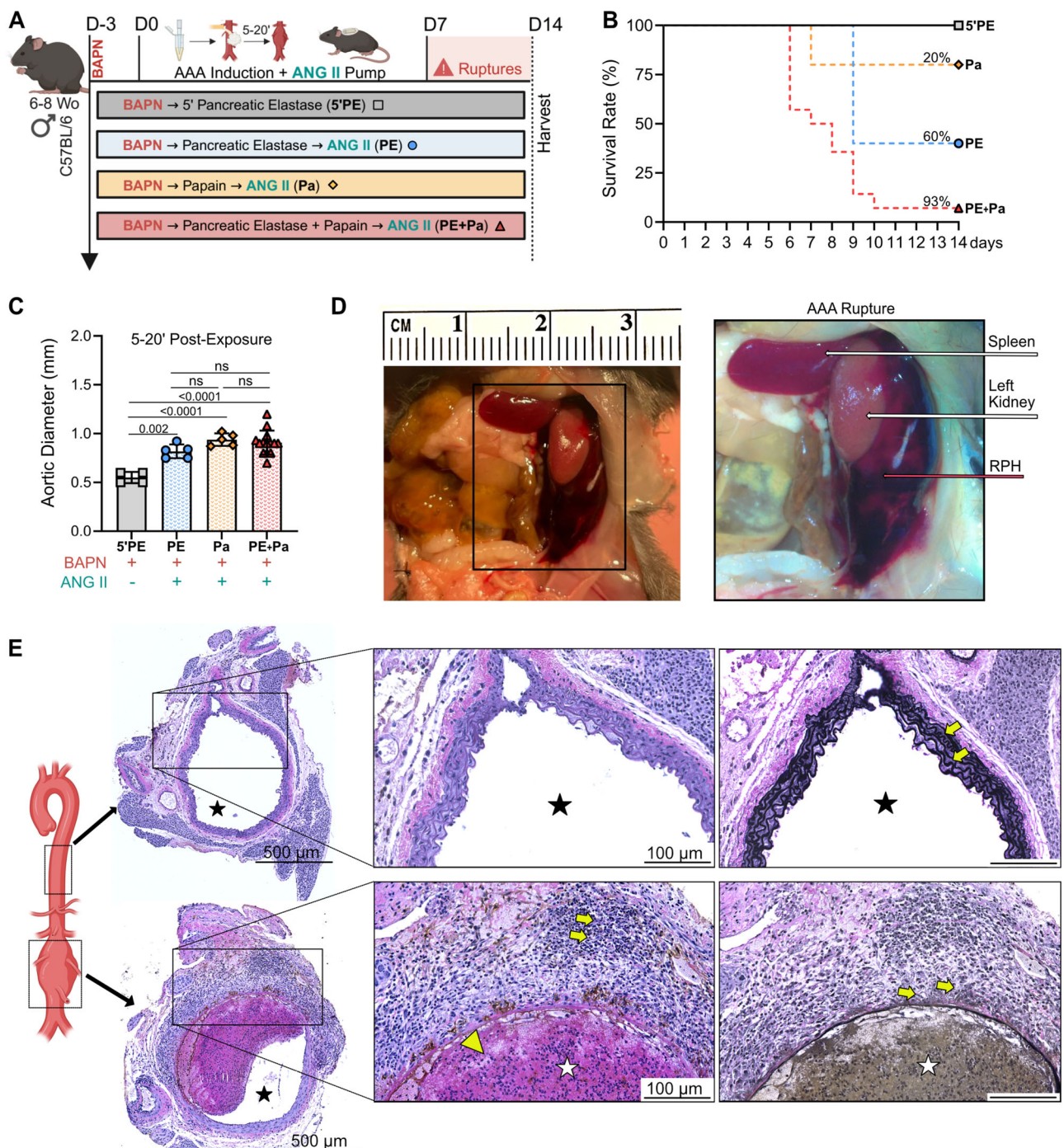

**Fig. 3 | Angiotensin II amplifies infrarenal AAA rupture, particularly in the PE + Pa combination model. A** Mice underwent either 5 min topical exposure to pancreatic elastase (5'PE, *n* = 4) or a 20-minute exposure to pancreatic elastase (PE, *n* = 5), papain (Pa, *n* = 5), or the combination (PE+Pa, *n* = 14). All mice were administered BAPN starting 3 days prior to the initiation of aneurysm creation. Additionally, the PE, Pa and PE+Pa groups received an ANG II subcutaneous pump to induce AAA rupture. Surviving mice were harvested on day 14 for histological assessment. Panel A was made using BioRender.com. **B** Kaplan-Meier curve demonstrating rate of survival following AAA induction to demonstrate infrarenal AAA rupture events. **C** Aortic diameter evaluation in millimeters at day 0 (either 5- or 20 min post-exposure). Aneurysms were defined as a diameter greater than 1 mm (a 50% increase from baseline measurements). **D** Representative AAA rupture into the left retroperitoneum, associated organs and retroperitoneal hematoma identified with the arrows. **E** H&E and VVG staining of the thoracic and abdominal aortas (cross-section of tissue slides) with 5x and 20x magnification. The star indicates the lumen, yellow arrows indicate cell infiltration and elastin fibers, and the large yellow arrow indicates intraluminal thrombus formation. RPH = Retroperitoneal Hematoma. Data are presented as mean ± standard deviation (SD). Ns>0.05, and statistical analysis was performed using one-way ANOVA for aortic diameter and Log rank test for survival rate.

**Table 1 | Group models of AAA Development and Rupture**

| Model | Group | Topical Exposure (µL) | Additional Exposure | Timeline (Days) | Ao Diameter (mm) Mean ± SD | Rupture Rate |
|---|---|---|---|---|---|---|
| AAA | **NS** | Normal Saline (50) | N/A | 14 | 0.5 ± 0.07 | 0% |
| | **PE** | Pancreatic Elastase (50) | N/A | 14 | 1.4 ± 0.3 | 0% |
| | **Pa** | Papain (50) | N/A | 14 | 1.8 ± 0.2 | 0% |
| | **PE + Pa** | Pancreatic Elastase (50) Papain (50) | N/A | 14 | 1.7 ± 0.3 | 0% |
| Chronic AAA | **WT** | No exposure | BAPN | 42 | 0.53 ± 0.06 | 0% |
| | **NS** | Normal Saline (50) | BAPN | 42 | 0.56 ± 0.02 | 0% |
| | **PE** | Pancreatic Elastase (50) | BAPN | 42 | 4.7 ± 1.2 | 20% |
| | **Pa** | Papain (50) | BAPN | 42 | 4.3 ± 1.3 | 0% |
| | **PE + Pa** | Pancreatic Elastase (50) Papain (50) | BAPN | 42 | 5.1 ± 0.7 | 17% |
| Rupture AAA | **5'PE** | Pancreatic Elastase (5)* | BAPN | 6 | 1.0 ± 0.4 | 0% |
| | | | | 14 | 1.4 ± 0.4 | |
| | **PE** | Pancreatic Elastase (50) | BAPN ANG II | 14 | N/A** | 60% |
| | **Pa** | Papain (50) | BAPN ANG II | 14 | N/A** | 20% |
| | **PE + Pa** | Pancreatic Elastase (50) Papain (50) | BAPN ANG II | 6 | 1.3 ± 0.2 | 93% |
| | | | | 14 | N/A** | |

*AAA* Abdominal Aortic Aneurysms, *NS* Normal Saline, *PE* Pancreatic Elastase, *Pa* Papain, *WT* Wildtype, *5'PE* 5 minutes Pancreatic Elastase, *N/A* Non-applicable or non-available, *BAPN* β-aminopropionitrile. *ANG II* Angiotensin II, *Ao* Aortic, *SD* Standard Deviation.
*5'PE is the only model with 5 minutes topical exposure without a cotton ball
** Only few survived and not enough to derive a reliable mean and SD

## Discussion

In summary, our study demonstrates that the combination of specific factors known to induce aneurysm growth exhibits a synergistic effect on AAA development by enhancing inflammation, MMP activation, elastin fiber degradation, and ILT formation – closely mirroring human disease characteristics[27,28]. First, we demonstrate that peri-adventitial exposure to a combination of PE+Pa is feasible and that it significantly influences AAA expansion, histopathology, and inflammation over a 14-day period. Secondly, when further enhanced by 0.3% BAPN administration over 42 days, PE+Pa combination led to the formation of a chronic AAA, similar to those in PE and Pa alone underscoring the model's utility for long-term studies and distinct inflammatory signals. Lastly, the ANG II pump addition, achieves a remarkable 93% rate of infrarenal AAA ruptures in the PE+Pa combination, which is significantly higher than when using PE or Pa alone. Additionally, PE+Pa combination exhibits a significant elevation in pro-inflammatory cytokines (e.g., MCP-1, IL-1β, IL-6, IL-17A) and MMP activity (MMP2 and MMP9), validating its relevance for investigating AAA progression and potential therapeutic interventions.

In humans, AAAs are most frequently fusiform and develop more often in the infrarenal abdominal aorta, with prior in vivo analysis demonstrating less organized collagen fibrils in human aortic tissue in this region[27–30]. As such, models recapitulating aneurysm formation in this anatomic region are crucial for understanding disease progression and modification. In our study, twenty minutes of peri-adventitial incubation with either PE, Pa and PE+Pa through a cotton ball, demonstrate a 100% rate of aneurysm development and growth localized to the infrarenal abdominal aorta, with a predominately fusiform morphology (Supplementary Fig. 1 and Fig. 2D). Additionally, aneurysm ruptures occur exclusively in the infrarenal abdominal aorta (Fig. 3D). This precise localization is critically important as it mirrors the predominant site of AAA formation and rupture in humans[29,30], underscoring the translational relevance and accuracy of our model.

The widely adopted ANG II infusion technique in *Apoe*⁻/⁻ mice, has been repeatedly demonstrated to lead to aortic ruptures[16]. While this model replicates hemodynamic and inflammatory changes associated with human AAA and achieves up to a 70% rupture rate when combined with 0.2% BAPN, a limitation of the traditional technique is that aortic ruptures often occur in the thoracic aorta rather than the infrarenal location as seen in the majority of human disease patterns. Also, aneurysms that occur from this model typically result from aortic dissections rather than de novo AAAs[16,17]. This limits the potential translation to human disease pathology and detailed mechanistic/therapeutic investigations. In contrast, our study demonstrates that combining a novel PE+Pa topical chemical exposure with ANG II infusion and BAPN administration results in significantly higher rupture rates. Importantly, the aneurysms in our model are uniquely localized to the infrarenal abdominal aorta, with no dissection or tears observed in the thoracic aorta (Supplementary Fig. 3). Therefore, the specificity of our model to the infrarenal abdominal aorta provides a more representative and focused approach to studying AAA pathology and interventions.

Moreover, the peri-adventitial elastase model is another widely adopted method that reliably induces infrarenal AAAs and achieves rupture with the addition of 0.2% BAPN[14,19]. While this model shows consistency and chronicity in AAA development, it falls short in rupture rates, reaching a maximum of 31% in advanced stages[19]. By replicating this model, we demonstrated a clear superiority of our PE+Pa combined with BAPN and ANG II (Fig. 3), which demonstrated significantly higher rates of rupture. Additionally, our model showed significantly elevated tissue inflammatory markers, including MCP-1, IL-6, and MMP2 and MMP9 activities – markers extensively linked to AAA disease and particularly prominent in AAA rupture. These markers were significantly elevated, just prior to rupture (Fig. 4B, E & H–J). The novel combination of pancreatic elastase and papain, with a 20 min abdominal aorta incubation, underscores the efficacy in enhancing AAA progression and rupture pathophysiology[28]. Our findings highlight the enhanced efficacy of our model in mimicking human AAA rupture pathophysiology, crucial for investigating interventions to mitigate rupture risk.

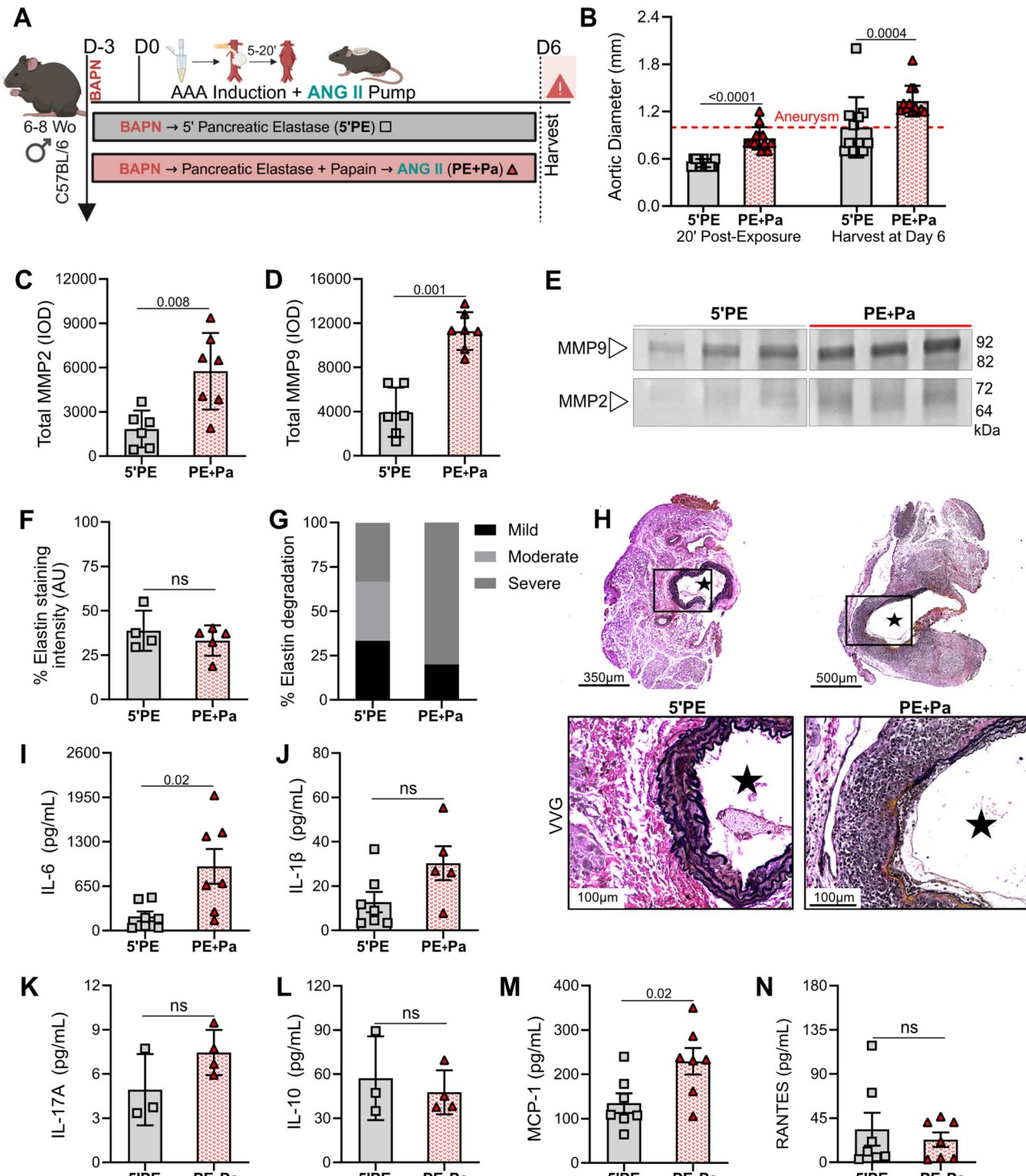

**Fig. 4 | Impact of inflammation and matrix metalloproteinases at day 6 in PE + Pa. A** Mice underwent either 5-minute topical exposure to pancreatic elastase (5'PE, $n = 11$) or 20-minute exposure to the PE and Pa combination (PE+Pa, $n = 13$). Additionally, all groups received BAPN on drinking water, whereas PE+Pa received ANG II subcutaneous pump. To assess AAA inflammation and matrix metalloproteinases prior to AAA rupture, day 6 was selected as the time of harvest. Mice from 5'PE and PE+Pa groups ($n = 7$) were processed for protein analysis. Panel A was made using BioRender.com. Chemokines (**B**) MCP-1 and (**C**) RANTES and pro-inflammatory markers (**D**) IL-1β, (**E**) IL-6, (**F**) IL-17A and (**G**) IL-10 content within the AAA tissue measured by ELISA. 2-3 tissue samples from PE+Pa and 4

from 5'PE were excluded from IL-1β, IL-17A and IL-10 measurements due to insufficient or undetectable readings. **H** Quantification of total MMP-2 and (**I**) total MMP-9 levels via integrated density (IOD). **J** Zymogram demonstrating total MMP-9 and MMP-2 activity bands. One tissue sample from 5'PE was excluded due insufficient space on the zymogram gel. Mice from the 5'PE group ($n = 4$) and PE+Pa group ($n = 5$) were harvested and processed for histopathological analysis, one mouse from the PE+Pa group died prior to tissue collection at day 6. **K** VVG staining, (**L**) quantification and (**M**) qualitative analysis of the abdominal aortas (cross-section of tissue slides) with 5x and 20x magnification. Star = lumen. Ns>0.05, and statistical analysis was performed using two tailed Mann-Whitney-test.

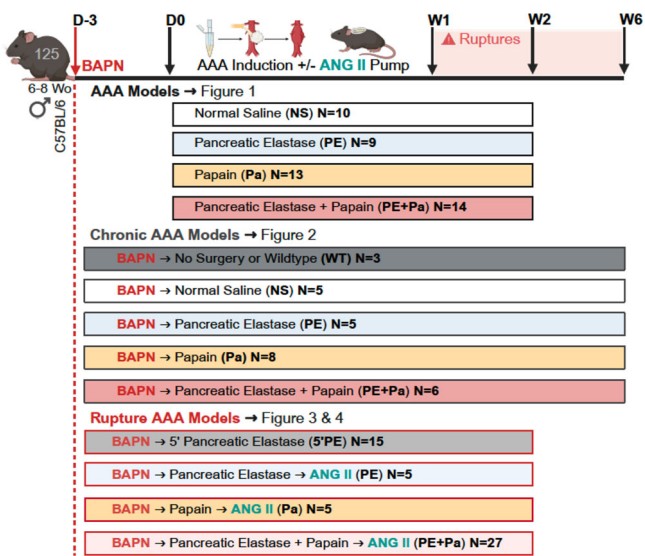

**Fig. 5 | Summary of the methodology to model development to study AAA creation and rupture.** Figure was made using BioRender.com.

aligning with previously reported data on their role in AAA[37]. The inflammatory milieu observed in our model closely mirrors that seen in human AAA disease[34], reinforcing the relevance of the models.

Several limitations must be acknowledged in our study, which may impact the scope of our findings. Firstly, our study exclusively used male mice. Although AAA prevalence is higher in males, including both sexes in future studies will be essential to assess sex-specific mechanisms and responses in AAA development and rupture. Secondly, the PE+Pa model exhibited a notably high and rapid rupture rate (93%), which might not fully represent the chronic progression of AAA in humans. While this acute rupture model offers valuable insights into mechanisms leading to rupture, the chronic AAA model will provide complementary information related to the slower and more progressive development of AAA. Additionally, our study did not employ $Apoe^{-/-}$ mice, typically used to investigate AAAs in the context of atherosclerotic disease. However, applying our models with in $Apoe^{-/-}$ mice as well as other specific genetic knockins and knockouts may be vital for future AAA mechanistic investigations. In this context longitudinal monitoring of blood pressure and aortic diameter will be important to facilitate a detailed understanding of the hemodynamic and morphological changes associated with AAA progression. Despite this, the progression of AAA growth was consistent, and the impact of the ANG II infusion on blood pressure has been previously well reported[16].

## Conclusions

Here we present a novel model that offers a reliable framework for studying AAA rupture in mice and a robust platform for chronic assessment of AAA growth. This model replicates key human AAA characteristics, such as intraluminal thrombus formation and elevated inflammatory markers, enabling reliable measurements of AAA expansion with minimal technical and procedural complications. It is a valuable tool for both acute and chronic AAA mechanistic research. The enhanced inflammatory response and matrix metalloproteinase activities we observe underscores its efficacy in mimicking rupture pathophysiology. This specificity to the infrarenal abdominal aorta, coupled with significant elevations in cytokines like MCP-1, IL-6, IL-1β, and MMP2 and MMP9, facilitates detailed investigations into AAA progression and potential therapeutic interventions. Additionally, the chronic model we report provides insights into long-term disease mechanisms, revealing differences in inflammatory signals compared to acute rupture models.

## Data availability

The datasets generated and/or analyzed during the current study are available from the corresponding author on reasonable request. See Supplementary Data for all source data, raw numerical results, and original zymography gels for MMP2 and MMP9.

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

Lastly, among the spectrum of models designed to induce infrarenal AAA rupture, the use of Transforming Growth Factor-beta (TGF-β) blockade represents another interesting and promising approach[31]. This model achieves a rupture rate of 40% by inhibiting the TGF-β pathway, which plays a critical role in tissue homeostasis and the inflammatory response. Its potential to reliably mimic AAA in humans is significant. However, adoption by the research community has been limited due to several factors[31]. Firstly, the high cost associated with in vivo TGF-β antibody infusion presents a substantial barrier to widespread use. Secondly, blocking TGF-β – a pathway integral to numerous physiologic cellular processes – raises concerns about its impact on the investigation of therapeutic mechanisms, which may suffer dilution of the effects pertaining to AAA rupture. Lastly, TGF-β mutations are found only in a specific subset of AAA patients and do not represent the broader AAA population. Despite these challenges, the model shows a notable capacity for replicating human AAA conditions, but with complexities that could interfere with comprehensive mechanistic studies. In contrast, our PE+Pa demonstrates a superior rupture rate of 93%, which is at least 53% higher relative to TGF-β. This significant increase in planned AAA rupture underscores the efficiency and robustness of the model. By avoiding the complexity associated with TGF-β antibody treatment, our approach also allows for a more straightforward interpretation of inflammatory responses and mechanical integrity within the aortic wall.

Another important strength of the PE+Pa is the presence of ILT formation (Supplementary Fig. 4), a distinctive feature of human AAAs that occurs in approximately 70–80% of patients. Recent studies evaluating human AAA biomechanics suggested ILT has a protective role against rupture by reducing strain imposed on the aortic wall[32,33]. Both of our rupture models exhibited ILT formation (Figs. 2 and 3) especially our chronic model, facilitating detailed investigations into its role in aneurysm disease progression. ILT is known to contribute significantly to AAA pathology, providing a relevant and translational aspect to our newly characterized murine models.

Our model also enabled reliable measurements of various cytokines associated with inflammation in AAA tissue, which are typically linked to disease severity in humans. Inflammatory cytokines, such as MCP-1, IL-6, and IL-1β, play crucial roles in aneurysm tissue and circulation[34,35]. Recently, our group demonstrated significantly elevated levels of MCP-1, underscoring the importance of the CCR2-MCP1 axis in AAA rupture and highlighting the potential of this pathway for therapeutic investigation[36]. Additionally, MMP2 and MMP9, were markedly increased in our model,

6. Potteaux, S. & Tedgui, A. Monocytes, Macrophages and Other Inflammatory Mediators of Abdominal Aortic Aneurysm. *Curr. Pharm. Des.* **21**, 4007–4015 (2015).

7. Li, H. et al. Modulation of immune-inflammatory responses in abdominal aortic aneurysm: Emerging molecular targets. *J. Immunol. Res.* **2018**, 7213760 (2018).

8. Ijaz, T., Tilton, R. G. & Brasier, A. R. Cytokine amplification and macrophage effector functions in aortic inflammation and abdominal aortic aneurysm formation. *J. Thorac. Dis.* **8**, E746–E754 (2016).

9. Dale, M. A., Ruhlman, M. K. & Baxter, B. T. Inflammatory cell phenotypes in AAAs: Their role and potential as targets for therapy. *Arterioscler Thromb. Vasc. Biol.* **35**, 1746–1755 (2015).

10. Hirsch, A. T. et al. ACC/AHA 2005 practice guidelines for the management of patients with peripheral arterial disease (Lower extremity, renal, mesenteric, and abdominal aortic). *Circulation* **113**, e463–e654 (2006).

11. Daugherty, A. & Cassis, L. A. Mouse Models of Abdominal Aortic Aneurysms. *Arterioscler Thromb. Vasc. Biol.* **24**, 429–434 (2004).

12. Yin, L., Kent, E. W. & Wang, B. Progress in Murine Models of Ruptured Abdominal Aortic Aneurysm. *Front Cardiovasc Med*. **12**, 950018 (2022).

13. Krishna, S. M., Morton, S. K., Li, J. & Golledge, J. Risk factors and mouse models of abdominal aortic aneurysm rupture. *Int. J. Mol. Sci.* **21**, 1–17 (2020).

14. Bhamidipati, C. M. et al. Development of a novel murine model of aortic aneurysms using peri-adventitial elastase. *Surg. (U. S.)* **152**, 238–246 (2012).

15. English, S. J. et al. Increased 18f-fdg uptake is predictive of rupture in a novel rat abdominal aortic aneurysm rupture model. *Ann. Surg.* **261**, 395–404 (2015).

16. Daugherty, A., Manning, M. W. & Cassis, L. A. Angiotensin II promotes atherosclerotic lesions and aneurysms in apolipoprotein E-deficient mice. *J. Clin. Invest.* **105**, 1605–1612 (2000).

17. Fashandi, A. Z. et al. A novel reproducible model of aortic aneurysm rupture. *Surg. (U. S.)* **163**, 397–403 (2018).

18. Lin, Y. C. et al. Mouse Abdominal Aortic Aneurysm Model Induced by Periarterial Incubation of Papain. *Lab Invest.* **103**, 100035 (2023).

19. Lu, G. et al. A novel chronic advanced stage abdominal aortic aneurysm murine model. *J. Vasc. Surg.* **66**, 232–242.e4 (2017).

20. Chao, C. L. et al. Advances and challenges in regenerative therapies for abdominal aortic aneurysm. *Front Cardiovasc Med*. **11**, 1369785 (2024).

21. Lysgaard Poulsen, J., Stubbe, J. & Lindholt, J. S. Animal Models Used to Explore Abdominal Aortic Aneurysms: A Systematic Review. *Eur. J. Vasc. Endovasc. Surg.* **52**, 487–499 (2016).

22. Sastriques-Dunlop, S. et al. Ketosis prevents abdominal aortic aneurysm rupture through C–C chemokine receptor type 2 downregulation and enhanced extracellular matrix balance. *Sci. Rep.* **14**, 1438 (2024).

23. Elizondo-Benedetto, S. et al. Chemokine Receptor 2 Is A Theranostic Biomarker for Abdominal Aortic Aneurysms. *JACC Basic Transl. Sci.* **10**, 101250 (2025).

24. Vandooren, J., Van Den Steen, P. E. & Opdenakker, G. Biochemistry and molecular biology of gelatinase B or matrix metalloproteinase-9 (MMP-9): The next decade. *Crit. Rev. Biochem Mol. Biol.* **48**, 222–272 (2013).

25. Franklin, M. K. et al. β-aminopropionitrile Induces Distinct Pathologies in the Ascending and Descending Thoracic Aortic Regions of Mice. *Arterioscler Thromb. Vasc. Biol.* **44**, 1555–1569 (2024).

26. Twenty Years of Studying Ang II (Angiotensin II)-Induced Abdominal Aortic Pathologies in Mice Continuing Questions and Challenges to Provide Insight Into the Human Disease. *Arterioscler Thromb. Vasc. Biol.* **42**, 277–288 (2022).

27. Lindeman, J. H. The pathophysiologic basis of abdominal aortic aneurysm progression: A critical appraisal. *Expert Rev. Cardiovascular Ther.* **13**, 839–851 (2015).

28. Nordon, I. M., Hinchliffe, R. J., Loftus, I. M. & Thompson, M. M. Pathophysiology and epidemiology of abdominal aortic aneurysms. *Nat. Rev. Cardiol.* **8**, 92–102 (2011).

29. De Freitas, S., D'Ambrosio, N. & Fatima, J. Infrarenal Abdominal Aortic Aneurysm. *Surgical Clin. North Am.* **103**, 595–614 (2023).

30. Liyanage, L. et al. Multimodal Structural Analysis of the Human Aorta: From Valve to Bifurcation. *Eur. J. Vasc. Endovasc. Surg.* **63**, 721–730 (2022).

31. Lareyre, F. et al. TGFβ (transforming growth factor-β) blockade induces a human-like disease in a nondissecting mouse model of abdominal aortic aneurysm. *Arterioscler Thromb. Vasc. Biol.* **37**, 2171–2181 (2017).

32. Thubrikar, M. J. Effect of thrombus on abdominal aortic aneurysm wall dilation and stress. *J. Cardiovasc. Surg.* **44**, 67–77 (2003).

33. Piechota-Polanczyk, A. et al. The Abdominal Aortic Aneurysm and Intraluminal Thrombus: Current Concepts of Development and Treatment. *Front Cardiovasc Med*. **2**, (2015).

34. Puchenkova, O. A. et al. Cytokines in Abdominal Aortic Aneurysm: Master Regulators With Clinical Application. *Biomarker Insights*. **17**, (2022).

35. Middleton, R. K. et al. The pro-inflammatory and chemotactic cytokine microenvironment of the abdominal aortic aneurysm wall: A protein array study. *J. Vasc. Surg.* **45**, 574–580 (2007).

36. Hafezi, S. et al. C-C Chemokine Receptor 2 is a Tissue Biomarker for Abdominal Aortic Aneurysmal and Occlusive Disease. *Ann Surg*. Epub ahead of print https://doi.org/10.1097/SLA. 0000000000006778 (2025).

37. Maguire, E. M., Pearce, S. W. A., Xiao, R., Oo, A. Y. & Xiao, Q. Matrix metalloproteinase in abdominal aortic aneurysm and aortic dissection. *Pharmaceuticals* **12**, 118 (2019).

## Acknowledgements

This work was supported by grants from National Institute of Health, National Heart Lung and Blood Institute R01HL153436 (Mohamed A. Zayed), R01HL150891 (Mohamed A. Zayed), R01HL153262 (Mohamed A. Zayed).

## Author contributions

Conceptualization: S.E.B., M.A.Z., M.S.Z; Methodology: B.A., S.E.B., M.S.Z.; Investigation: S.E.B., M.S.Z., I.K, R.W.; Data Collection: S.E.B., M.S.Z., I.K.; Supervision: M.A.Z.; Writing – original draft: S.E.B., M.A.Z.; Writing – review & editing: S.E.B., R.W., M.A.Z.

## Competing interests
The authors declare no competing interests.
