## [Transparent Peer Review file · Communications Medicine]

Synergistic Elastase and Papain Injury Drives Abdominal Aortic Aneurysm Formation and Rupture in Mice

Corresponding Author: Dr Mohamed Zayed

Version 0:

Reviewer comments:

Reviewer #1

(Remarks to the Author)

This study by Elizondo-Benedetto et al. investigates whether combining multiple known inducers of abdominal aortic aneurysm (AAA) in mice could lead to new models that recapitulate the chronic nature of human AAA disease better than existing models. The authors investigated in a step-wise fashion the addition of each pre-existing inducer to the peri-adventitial elastase model (PPE) and the resulting effect on the AAA development. This study tackles an important challenge in AAA research, due to the current mouse-to-human translatability issue of promising treatments. While the results of the study could contribute to the methodological advancement of the field, there are some issues that need addressing in order to make sure that others understand how the proposed models differ from each other and established ones, and most importantly, that the models can be reproduced by other labs.

Major Comments:

1. The title of the article does not fully reflect the findings of the publication. While "Papain-Elastase" is one of the models presented in this publication, it doesn't show any ruptures, it is only with the addition of BAPN that 17% rupture is reported (which is less than the reported 20% for the established PPE + BAPN). The proposed novel rupture model (NvRM), which leads to 93% rupture rate, doesn't only have "Papain-Elastase" but additionally BAPN and Ang-II. Please revise the title accordingly.
 - a. Additionally, especially in the discussion and conclusion, it was unclear at times which model was being discussed, please state the model by name and not just "our model" or "a novel model" or "chronic model" when there's multiple new models proposed in this study. For example, page 8, line 27 says "Our model" and the authors reference Figure 2 (chronic models, which there is multiple), then the line after the authors reference Figure 3 (NvRM). This is a recurring issue.
2. While the models used are nicely summarized in table 1 and the schematic figure 6, it is unclear why only certain comparison were performed when Angiotensin II was added. In Figure 1, PPE, PAP, and PPE+PAP are directly compared, and later, with BAPN, the same three groups are presented. However, once ANG II is introduced, the analysis is limited to PPE+BAPN and PPE+PAP+BAPN+ANGII, even though the earlier chronic models all showed comparable results.
 - a. Since the manuscript emphasizes the importance of combining inducers, comparison of PPE+BAPN+Ang II and PAP+BAPN+Ang II would seem important to assess whether combining PPE and PAP meaningfully contributes in this setting.
 - b. Similarly, inclusion of PPE + ANG II ± BAPN and PAP + ANG II ± BAPN groups would clarify whether the combined approach is necessary, or whether ANG II alone already produces the reported effects.
 - c. Since the paper presents many model combinations that could greatly benefit researchers, with table 1 presenting an already very helpful as a summary of models, extending it with available quantitative outcomes (e.g., elastin/collagen degradation, VSMC content, MMP2/MMP9) would make it an even more practical reference for researchers selecting models.
3. The NvRM model has an impressive rupture rate, and could potentially be very useful in research, but several aspects of its characterization remain unclear:
 - a. In the discussion (page 9, line 8-9) the authors note that ANG II models often produce aneurysms via dissection rather than de novo formation; however, no direct evidence is presented that the NvRM produces de novo AAAs. Early diameter measurements (day 6) would be particularly informative, since most animals ruptured by endpoint.
 - b. The claim that the "rupture models exhibited ILT formation" (page 10, line 18) is not supported by clear data. For day 6, please provide the frequency of ILT, intimal/medial changes, or dissections/intramural thrombus in the abdominal aorta.
 - c. The discussion (page 9, lines 24-28) suggests that PPE + PAP enhances rupture, but based on the data, rupture appears

to depend primarily on the addition of BAPN and BAPN + ANG II (17% and 93% rupture rates, respectively). Please revise this statement to more accurately reflect the findings.

d. Given the very rapid rupture kinetics of the NvRM, please discuss whether and how this model truly reflects human AAA rupture pathophysiology and chronic AAA progression, as stated in the conclusion.

e. The comparison to the TGF β model is interesting, but it is not clear why the use of four inducers simultaneously (including systemic agents such as BAPN and Ang II) would reduce, rather than increase, concerns about off-target effects. Please clarify this point.

4. For several models, the justification of concentrations used is unclear due to either missing references or lack of explanation. This information is critical for reproducibility and for other researchers to apply these models:

a. Was the PPE used at a 100% concentration? i.e. 50 μ L of 10.3 mg/mL? Please indicate in the methods. Additionally, page 12 line 18 in the methods says "As previously reported," to describe the PPE model, yet no study is referenced. The Bhamidipati et al. study referenced in line 14 did not use the cotton ball method and only 10 μ L elastase for 10 min incubation.

b. Similarly, line 28 on the same page again says "As previously described," with no reference, please reference appropriately.

c. In the methods under "Chronic model of AAA in male C57BL/6 mice", "0.3% BAPN in water" was used in this study, referencing a publication of an AAA rat model (ref 15), when the study that established this model used 0.2% BAPN (ref 31). Please clarify the reasoning for the increase in concentration in text or reference the publication with the protocol followed.

d. The angiotensin II was indicated to be used at 2000 ng/kg/min, but both studies referenced used ApoE KO mice. In fact, 1000 ng/kg/min is the typical in the ApoE KO model as established by Daugherty et al. (ref 25). Please explain the reasoning behind the concentration used or reference the publication with the protocol the authors followed for C57BL/6 mice.

Minor Comments:

1. Figure 1 D, in the PPE model VVG: there is hardly any elastin degradation shows, please choose a more representative image that matches the quantification in Fig 1E

a. Also, in the same figure, some times ns is indicated while other times there is nothing indicated, which should also mean ns. Please stay consistent across panels to avoid confusion.

2. In the NvRM, it is unclear what type of ruptures occurred, were they all abdominal or were there any thoracic ones observed? Please indicate with AA ruptures for clarity. Also, in line 5, page 7, "NvRM" is used without clear indication of the acronym. Please write out "Novel rupture model (NvRM)" at first mention.

3. The elastase-induced model is referred to as PPE throughout most of the results, but in Fig. 4 and the related text, "E+B" appears. Please keep terminology consistent (e.g., PPE+BAPN) as elsewhere.

4. In the results, page 8, line 6, it says "elastin degradation was significantly decreased in both E+B and NvRM groups when compared to controls, but qualitative analysis demonstrated a more severe degradation in the NvRM group". Please double check for accuracy, since the figures appear to show more severe degradation, not "decreased".

5. In Supplementary figure 3B, the average weight of mice in PAP group appears to drop from week 3 to week 4 then recovers in week 5. Please comment on this. Additionally, please indicate the n numbers that comprise the data in Supp Fig 3, as well as Supp Fig 2 B-D, and Supp Fig 4.

6. In Supplementary figure 4, the legend doesn't specify which model the images are from, please indicate.

7. In the methods section, it is indicated that mice used in the study were 6-8 weeks old. Typically, wildtype mice used in elastase-induced AAA studies, including ones cited here, are 8-12 weeks old at AAA induction. Please comment to this.

8. In the methods, under "Chronic model of AAA in male C57BL/6 mice", it is indicated that "In addition to either Pap or PPE+Pap exposures, two groups of mice also underwent β -aminopropionitrile (BAPN) administration through drinking water (0.3% BAPN in water) to promote AAA rupture", however the publication also presents PPE + BAPN, as well as controls. Please update the information to be comprehensive of all the groups.

9. In Table 1, under the "Rupture AAA" model, the volume for elastase in "BAPN+E" is listed as "5," which appears to be a typo, likely 50 μ L. Please verify.

10. In Figure 6, The use of "C" in parentheses (e.g., "C BAPN," "C Pap") is unclear. If "C" stands for "Chronic," the placement before the inducer is confusing since BAPN is what results in the chronic aspect. Please clarify in the legend or remove.

Overall, this is an interesting and potentially valuable study. Addressing the points above will strengthen the manuscript and make it more accessible and reproducible for others in the field.

Reviewer #2

(Remarks to the Author)

The authors propose a novel mouse model of abdominal aortic aneurysm (AAA) that combines established induction methods to create either a chronically enlarged aneurysm model or an early rupture model. One of the major reasons why basic research on AAA treatment has not translated successfully into clinical practice is that pharmacological agents effective in mouse models often fail to show efficacy in humans. In particular, there are few reliable models for aneurysm rupture, and the model proposed by the authors could greatly advance our understanding of AAA pathophysiology.

In the chronic and rupture models, there appears to be no significant difference in the levels of the cytokines and proteolytic enzymes presented, suggesting that other factors may contribute to rupture. Please discuss this point further, including potential differences between mice and humans. If blood pressure is involved, please include blood pressure data as well.

In the chronic model, the aneurysm enlarged substantially, and the presence of intraluminal thrombus resembles findings

observed in human AAAs. Some mouse models show spontaneous regression of aneurysms over time, whereas in humans, aneurysms always continue to expand and eventually rupture. Does this model show aneurysm regression or rupture after more than six weeks of follow-up?

Figure 1C: The aneurysm appears to form acutely within 20 minutes after chemical exposure. This differs from human AAAs, which develop over a chronic course. How do the authors interpret this difference?

Page 4, line 26: The manuscript states that "aortic diameter continued to grow over a 14-day follow-up," but only the diameters at the time of chemical exposure and at day 14 are presented. It is therefore unclear whether the aneurysm enlarged rapidly or gradually. Please provide intermediate measurements to clarify the growth pattern.

How many mice were used in each experiment? Across all experiments, were there any mice that died during or after surgery from causes other than rupture?

Figure 2B: The number of mice appears to vary among groups. Is this due to deaths during the study? Were the aortic diameters of mice that died from rupture included in the analysis?

Figures 3F, 3G, Supplementary Figures 2B and 2D: How were the numbers of inflammatory cells and vascular smooth muscle cells quantified? Please describe in detail in the Methods section which staining method was used, which region of the tissue was analyzed, and under what microscopic field the counts were performed.

Figures 3 and 4: Data on aortic diameter at the time of rupture are missing. In human AAAs, rupture is rare when the diameter is less than twice normal, and the risk increases with further enlargement. Since rupture in this model may occur before significant aortic dilation, measurement of aortic diameter at rupture is essential.

Figures 1M and 4F: The y-axis values for IL-17A differ markedly between the two graphs.

Page 9, line 24: It cannot be concluded that inflammatory markers peak at day 6 based only on the data from that time point. Time-course data are needed to support this statement.

Version 1:

Reviewer comments:

Reviewer #1

(Remarks to the Author)

The revisions have greatly improved the quality of the manuscript and position the introduced models more clearly in comparison to existing ones. The tables in particular now provide a comprehensive comparison, demonstrating the great efforts the authors have undertaken.

Just a couple of minor comments:

-In Figure 3C, presenting the diameter at 5-20' post exposure which is the "acute initial dilation" is not clarifying AAA results. It would be more informative to replace this with 5/20' vs endpoint measurements like Figure 1C and 2B (if data is available), with the data of whichever mice survived at the endpoint even if statistical comparison isn't possible due to low n numbers (this can be denoted in the figure legend). This would give a more accurate picture of the AAA models' details, especially in the context of chronic AAA discussions.

-Discussion page 14 lines 403-405 "By replicating this model, we demonstrated a clear superiority of our PE+Pa (Fig. 3), which demonstrated significantly higher rates of rupture." This is an inaccurate comparison as BAPN+PE+Pa without AngII doesn't show a significantly higher rupture rate (17%) than BAPN+PE (20% in your study, 31% reported in original study); explicit mention of AngII is needed.

-In the abstracts results "Compared to previous models, the PE+Pa combination demonstrates an increase in rupture events..." this statement is similarly inaccurate since it's the BAPN+AngII+PE+Pa combination that does increase rupture rate to 93%, while BAPN+PE+Pa has the same rupture (17%) as BAPN+PE (20% in your study).

-Discussion page 15 lines 434-435 "Both our chronic and rupture models exhibited ILT formation (Fig. 2 and 3)," while the results are now clear for the chronic models, this overstates the findings for the rupture models since only a single surviving mouse in the rupture model showed ILT, while none were observed at day 6.

Overall, the authors have responded thoroughly and constructively to all my previous major and minor comments.

Reviewer #2

(Remarks to the Author)

The authors have largely addressed my questions, and the additional information regarding the rupture model has made the importance of this model much clearer. With respect to the newly added rupture data, I would like to raise an additional point. In humans, the risk of aneurysm rupture increases with aneurysm diameter, and aneurysm size is a key criterion for clinical

intervention. Because aneurysms typically rupture only after sufficient enlargement in humans, the aortic diameter at the time of rupture is clinically highly relevant when mouse rupture models are used for research.

In Figure 3C, the aneurysm diameters measured in the PE + Pa group appear to show greater variability compared with the other groups. Was there an association between aneurysm diameter and the occurrence of rupture? If aneurysms with larger diameters or faster expansion rates are more prone to rupture, this would further support the translational relevance of this model to human AAA disease.

Point-by-point response to reviewer's comments:

We appreciate the thorough review provided by the reviewers and the editor. The suggestions and detailed feedback have significantly enhanced the quality of our original manuscript, allowing us to present the data more effectively and improve the reproducibility of our models of AAA development and rupture. Before addressing each individual response, we would like to clarify the **new nomenclature and abbreviations** for our tested groups. This change aims to increase reproducibility and understanding, and to address any confusions raised by the reviewers:

Previous	Revised
Wildtype (WT)	Wildtype (WT)
Saline	Normal Saline (NS)
Porcine Pancreatic Elastase (PPE)	Pancreatic Elastase (PE)
Papain (Pap)	Papain (Pa)
Porcine Pancreatic Elastase + Papain (P+P)	Pancreatic Elastase + Papain (PE+Pa)
Novel Rupture Model (NvRM)	PE+Pa (with the addition of BAPN and ANG II)
Elastase + BAPN (E+B)	5 minutes Pancreatic Elastase (5'PE)
If BAPN is included to the model, it will be indicated in the figure timeline and/or graphs. If ANG II is included to the model, it will also be indicated in the figure timeline and/or graphs.	

Reviewer #1 (Remarks to the Author):

This study by Elizondo-Benedetto et al. investigates whether combining multiple known inducers of abdominal aortic aneurysm (AAA) in mice could lead to new models that recapitulate the chronic nature of human AAA disease better than existing models. The authors investigated in a step-wise fashion the addition of each pre-existing inducer to the peri-adventitial elastase model (PPE) and the resulting effect on the AAA development. This study tackles an important challenge in AAA research, due to the current mouse-to-human translatability issue of promising treatments. While the results of the study could contribute to the methodological advancement of the field, there are some issues that need addressing in order to make sure that others understand how the proposed models differ from each other and established ones, and most importantly, that the models can be reproduced by other labs.

Major Comments:

1. The **title of the article** does not fully reflect the findings of the publication. While "Papain-Elastase" is one of the models presented in this publication, it doesn't show any ruptures, it is only with the addition of BAPN that 17% rupture is reported (which is less than the reported 20% for the established PPE + BAPN). The proposed novel rupture model (NvRM), which leads to 93% rupture rate, doesn't only have "Papain-Elastase" but additionally BAPN and Ang-II. Please revise the title accordingly.

We appreciate the reviewer's insightful comment and value the suggestion. We agree that the title should more accurately reflect the novelty of our model. To clarify, the true innovation in our study lies in the combination of pancreatic elastase and papain, rather than the individual use of these enzymes. This unique combination (PE+Pa) has proven to be the most efficient chemical incubation method for the consistent development of AAA. It facilitates elastin and collagen degradation, promotes MMP activity, and significantly elevates key pro-inflammatory cytokines and chemokines strongly associated with AAA disease severity (**Fig.1**).

With the addition of BAPN, we observed the highest AAA growth rate, second highest rupture rate and substantial histological severity (**Fig. 2**). Furthermore, the introduction of ANG II demonstrated the most efficient rupture kinetics (93%) compared to the use of pancreatic elastase and papain alone, as well as the greatest postoperative recovery, as detailed in the revised manuscript (**Fig.3**). This made us continue our investigation with this unique combination and, when compared with the widely used 5-minute topical pancreatic elastase (5'PE) rupture model, PE+Pa demonstrated a significantly higher rupture rate (**Fig. 3B**), MMP activation, inflammation and disease severity at day 6 harvest (**Fig. 4**). As the reviewer pointed out, this step-wise study elucidated the best option for investigating AAA

progression and rupture. It also opens up new avenues for future studies using this combination, which is both feasible and reproducible.

We understand that this may not have been evident before, but thanks to the reviewers' concerns, we supported our findings with a new study. The follow-up investigation clearly highlighted the importance of this combination. Moreover, while we acknowledge that the addition of BAPN and Ang-II further enhances the rupture rates, the key novelty remains the synergistic effect of the Pancreatic elastase and papain combination (PE+Pa).

To address the reviewer's comments and the editor's request to ensure the title is concise yet comprehensive (less than 15 words), we have revised it to:

"Synergistic Elastase and Papain Injury Drives Abdominal Aortic Aneurysm Formation and Rupture in Mice"

a. Additionally, especially in the discussion and conclusion, it was unclear at times which model was being discussed, please state the model by name and not just "our model" or "a novel model" or "chronic model" when there's multiple new models proposed in this study. For example, page 8, line 27 says "Our model" and the authors reference Figure 2 (chronic models, which there is multiple), then the line after the authors reference Figure 3 (NvRM). This is a recurring issue.

We appreciate the reviewer's comments and valuable recommendations. We absolutely agree that our manuscript demonstrates several potential models, which should be better summarized to guide readers in selecting the most appropriate one. As discussed earlier, we have updated the **nomenclature and abbreviations** to address this recurring issue:

Previous	Revised
Wildtype (WT)	Wildtype (WT)
Saline	Normal Saline (NS)
Porcine Pancreatic Elastase (PPE)	Pancreatic Elastase (PE)
Papain (Pap)	Papain (Pa)
Porcine Pancreatic Elastase + Papain (P+P)	Pancreatic Elastase + Papain (PE+Pa)
Elastase + BAPN (E+B)	5 minutes Pancreatic Elastase (5'PE)
If BAPN is included to the model, it will be indicated in the figure timeline and/or graphs. If ANG II is included to the model, it will also be indicated in the figure timeline and/or graphs.	

Additionally, we have clarified the text to specify whether it refers to the chemical incubation alone or the addition of BAPN and/or ANG II when discussing each model individually. Given that the main differences between **Fig. 1, 2, and 3/4** are either the

addition of BAPN or BAPN + ANG II, we have decided to use the name of the topical agent and then clarify if it pertains to the BAPN-only model or the BAPN-ANG II model.

The term "Novel Rupture Model (NvRM)" has been removed and replaced with the PE+Pa combination with the addition of BAPN and ANG II. This correction should resolve the concern, and we hope the reviewer agrees with this approach. Here is the updated **Fig. 5** with all models and names, which we have included in our main manuscript and updated in **Table 1**.

Figure 5. Summary of the methodology to model development to study AAA creation and rupture. Figure was made using BioRender.com.

2. While the models used are nicely summarized in table 1 and the schematic figure 6, it is unclear why only certain comparison were performed when Angiotensin II was added. In Figure 1, PPE, PAP, and PPE+PAP are directly compared, and later, with BAPN, the same three groups are presented. However, once ANG II is introduced, the analysis is limited to

PPE+BAPN and PPE+PAP+BAPN+ANGII, even though the earlier chronic models all showed comparable results.

a. Since the manuscript emphasizes the importance of combining inducers, comparison of PPE+BAPN+Ang II and PAP+BAPN+Ang II would seem important to assess whether combining PPE and PAP meaningfully contributes in this setting.

We appreciate the reviewer's great suggestion and decided to perform this crucial experiment. Please refer to the revised result section **page 11, lines 18-29**:

"Infrarenal AAA rupture was further evaluated following BAPN administration, this time with the addition of ANG II subcutaneous pumps to induce aneurysm expansion and aortic instability.¹⁻³ Once again, mice underwent a 20-minute topical exposure to either Pa, PE or the PE+Pa combination. To evaluate rupture kinetics, a previously described model was replicated using 5 μ L of PE for a 5-minute topical exposure (5'PE) followed by BAPN administration (**Fig. 3A** and **Table 1**)⁴. Mice subjected to the PE+Pa combination demonstrate the highest rupture rate of 93% (13 out of 14 mice), with an average rupture time of 7.4 ± 1.5 days post-AAA induction (**Fig. 3B**). This is significantly higher when compared to the 5'PE model ($p=0.003$), which demonstrate 0% ruptures. Furthermore, the PE+Pa combination also leads to a significant increase in rupture events, with an absolute risk augmentation of 73% and 33% when compared to Pa and PE alone ($p=0.009$ and $p=0.03$ respectively), despite exhibiting similar post-exposure aortic diameter (**Fig. 3C**)."

We hope that, with the addition of this cohort, we have successfully addressed the reviewer's concerns about the synergistic effect or validity of PE and Pa as a combination.

b. Similarly, inclusion of PPE + ANG II \pm BAPN and PAP + ANG II \pm BAPN groups would clarify whether the combined approach is necessary, or whether ANG II alone already produces the reported effects.

We appreciate the reviewer's comment and believe we have clarified the necessity of the combined PE+Pa approach based on our findings and response in question (a).

While the addition of ANG II alone is an interesting idea, we it falls outside the scope of this manuscript, since it is a well-known traditional model and does not offer significant new insights. BAPN and ANG II are well-known promoters of the disease and play a crucial role in establishing the rupture model (but not necessarily limited to the infrarenal aorta). Our most remarkable finding is the unique synergistic effect of the PE+Pa combination compared to PE and Pa alone. To illustrate this, in **Fig. 3** we demonstrated that the addition of ANG II to the PE+Pa combination resulted in the highest rupture rate, despite similar aneurysm diameters when compared to all other groups tested. However, when the same

conditions were applied to the PE and Pa alone groups, the increase in rupture rate was not statistically significant when compared to the 5'PE model that did not receive ANG II: "However, the rupture events for PE and Pa are not statistically significant when compared to 5'PE." (Page 11, line 31 and page 12, line 1).

Here is the data for PE and Pa when BAPN and ANG II were included (Fig. 3B):

- PE (BAPN+ANG II) vs 5'PE: rupture rate of 60% vs 0%, p = 0.076
- Pa (BAPN+ANG II) vs 5'PE: rupture rate of 20% vs 0%, p = 0.8

Moreover, when PE and Pa groups that received both BAPN and ANG II were compared to the PE and Pa groups that only received BAPN (Fig. 2B), they demonstrated higher rupture events, but these increases were not statistically significant:

- PE+BAPN+ANG II vs PE+BAPN: rupture rate of 60% vs 20%, p = 0.171
- Pa+BAPN+ANG II vs Pa+BAPN: rupture rate of 20% vs 0%, p = 0.206

Thus, in the case of PE and Pa alone, the addition of ANG II was not sufficient to demonstrate a significant impact on AAA rupture. On the other hand, the PE+Pa combination with the addition of BAPN and ANG II did show a significant increase in rupture events compared to BAPN alone:

- (PE+Pa) +BAPN+ANG II vs (PE+Pa) +BAPN (rupture rate of 93% vs 17%, p = 0.0007).

Additionally, previous studies have shown that the addition of ANG II to well-established models enhances rupture rates, which aligns with our findings.⁵

c. Since the paper presents many model combinations that could greatly benefit researchers, with table 1 presenting an already very helpful as a summary of models, extending it with available quantitative outcomes (e.g., elastin/collagen degradation, VSMC content, MMP2/MMP9) would make it an even more practical reference for researchers selecting models.

We appreciate the reviewer suggestion regarding the table. We have updated **Table 1** to include the new groups performed and we have created a new **Supplementary Table 2**, with extended information for readers to use when selecting the desired model.

Table 1. Group models of AAA Development and Rupture

Model	Group	Topical Exposure (μL)	Additional Exposure	Timeline (Days)	Ao Diameter (mm) Mean ± SD	Rupture Rate
AAA	NS	Normal Saline (50)	N/A	14	0.5 ± 0.07	0%
	PE	Pancreatic Elastase (50)	N/A	14	1.4 ± 0.3	0%
	Pa	Papain (50)	N/A	14	1.8 ± 0.2	0%
	PE+Pa	Pancreatic Elastase (50) Papain (50)	N/A	14	1.7 ± 0.3	0%
Chronic AAA	WT	No exposure	BAPN	42	0.53 ± 0.06	0%
	NS	Normal Saline (50)	BAPN	42	0.56 ± 0.02	0%
	PE	Pancreatic Elastase (50)	BAPN	42	4.7 ± 1.2	20%
	Pa	Papain (50)	BAPN	42	4.3 ± 1.3	0%
	PE+Pa	Pancreatic Elastase (50) Papain (50)	BAPN	42	5.1 ± 0.7	17%
Rupture AAA	5'PE	Pancreatic Elastase (5)*	BAPN	6	1.0 ± 0.4	0%
				14	1.4 ± 0.4	
	PE	Pancreatic Elastase (50)	BAPN ANG II	14	N/A**	60%
	Pa	Papain (50)	BAPN ANG II	14	N/A**	20%
	PE+Pa	Pancreatic Elastase (50) Papain (50)	BAPN ANG II	6	1.3 ± 0.2	93%
				14	N/A**	

Abbreviations: AAA, Abdominal Aortic Aneurysms. NS, Normal Saline. PE, Pancreatic Elastase. Pa, Papain. WT, Wildtype. 5'PE, 5 minutes Pancreatic Elastase. N/A, Non-applicable or non-available. BAPN, β-aminopropionitrile. ANG II, Angiotensin II. Ao, Aortic. SD, Standard Deviation.
*5'PE is the only model with 5 minutes topical exposure without a cotton ball
** Only few survived and not enough to derive a reliable mean and SD

Supplementary Table 2. Group models of AAA Development and Rupture (Extended Information)

	Group: Topical Exposure (μL) Additional Exposure	Days	Ao Diameter (mm) Mean ± SD		ECM Degradation		VSMC Loss	Inflamm ation	ILT (%)	Cytokines	Chemokines	MMP9	MMP2	Rupture Rate (%)
			Post- exposure	Harvest	Elastin	Collagen				IL-6, IL- 1β, IL17A	MCP-1, RANTES			
AAA	NS: Saline (50)	14	0.5 ± 0.05	0.5 ± 0.07	N	N	Mild	Mild	0%	N	N	N	N	0%
	PE: Pancreatic Elastase (50)	14	0.9 ± 0.13	1.4 ± 0.3	↑↑	↑	Mild	Moderate	0%	↑	↑	↑	N	0%
	Pa: Papain (50)	14	1.0 ± 0.2	1.8 ± 0.2	↑↑↑	↑	Moderate	Moderate	0%	N	↑↑	↑↑	N	0%
	PE+Pa: PE (50) / Pa (50)	14	1.0 ± 0.07	1.7 ± 0.3	↑↑↑	↑	Moderate	Severe	0%	↑↑	↑↑↑	↑	N	0%
Chronic AAA	WT: No exposure BAPN	42	N/A	0.53 ± 0.1	N	N	Mild	Mild	0%	N/A	N/A	N/A	N/A	0%
	NS: Saline (50) BAPN	42	0.5 ± 0	0.56 ± 0.0	N	N	Mild	Mild	0%	N/A	N/A	N/A	N/A	0%
	PE: Pancreatic Elastase (50) BAPN	42	0.8 ± 0.05	4.7 ± 1.2	↑↑↑	↑↑	Severe	Moderate	50%	N/A	N/A	N/A	N/A	20%
	Pa: Papain (50) BAPN	42	1.15 ± 0.3	4.3 ± 1.3	↑↑↑	↑↑↑	Severe	Moderate	50%	N/A	N/A	N/A	N/A	0%
	PE+Pa: PE (50) / Pa (50) BAPN	42	1.3 ± 0.08	5.1 ± 0.7	↑↑↑	↑↑↑	Severe	Moderate	25%	N/A	N/A	N/A	N/A	17%
Rupture AAA	5'PE: Pancreatic Elastase (5) * BAPN	14	0.5 ± 0.05	1.4 ± 0.4	N/A	N/A	N/A	N/A	N/A	N/A	N/A	N/A	N/A	0%
		6	0.5 ± 0.04	1.0 ± 0.4	↑↑				0%	↑	↑	N	N	
	PE: Pancreatic Elastase (50) BAPN + ANG II	14	0.8 ± 0.07	N/A	N/A	N/A	N/A	N/A	N/A	N/A	N/A	N/A	N/A	60%
	Pa: Papain (50) BAPN + ANG II	14	0.9 ± 0.06	N/A	N/A	N/A	N/A	N/A	N/A	N/A	N/A	N/A	N/A	20%
	PE+Pa: PE (50) / Pa (50) BAPN + ANG II	14	0.9 ± 0.1	N/A	N/A	N/A	N/A	N/A	N/A	N/A	N/A	N/A	N/A	93%
6		0.9 ± 0.1	1.3 ± 0.2	↑↑	0%				↑↑	↑↑↑	↑↑↑	↑↑		

*5'PE: is the only model with 5 minutes topical exposure **without** a cotton ball.

N/A Non-Applicable or Not Available

3. The NvRM model has an impressive rupture rate, and could potentially be very useful in research, but several aspects of its characterization remain unclear:

a. In the discussion (page 9, line 8-9) the authors note that ANG II models often produce aneurysms via dissection rather than de novo formation; however, no direct evidence is presented that the NvRM produces de novo AAAs. Early diameter measurements (day 6) would be particularly informative, since most animals ruptured by endpoint.

We agree with the reviewer's comment and have now provided the aortic diameter measurements at day 6, collected prior to harvest for the PE+Pa combination as well as the 5'PE group (**Fig. 4A&B**). Additionally, we have updated **Table 1** and **Supplementary Table 2** to reflect the exact values for the day 6 harvest.

b. The claim that the "rupture models exhibited ILT formation" (page 10, line 18) is not supported by clear data. For day 6, please provide the frequency of ILT, intimal/medial changes, or dissections/intramural thrombus in the abdominal aorta.

Thank you for the valuable comment. Our PE+Pa model, supplemented with BAPN and ANG II, has proven to be a highly effective model for inducing infrarenal AAA rupture. We were able to harvest only one aneurysm that survived until the day of harvest, in which we identified an intraluminal thrombus (ILT), as clearly shown in **Fig. 3E**. Unfortunately, due to the high rupture rate and the dynamics of the newly established kinetic models, we were unable to harvest the ruptured tissue for further characterization of ILT formation and assess the histological severity of the ruptured tissue. Nevertheless, we re-analyzed our histological samples harvested at day 6, where we did not find any ILT formation (as shown in **Supplementary Table 2**). We did not observe dissection or intramural thrombus in the samples. In terms of intimal and medial changes, the PE+Pa group demonstrated classic medial expansion due to increased inflammation typically seen in AAA formation, when compared to the normal saline (NS) group (see representative figure below).

Representative cross-sectional histological images using hematoxylin and eosin (H&E) staining to assess aortic intimal and medial changes after AAA induction with either normal saline (NS) or the PE+Pa combination. The PE+Pa group demonstrates medial expansion

with inflammatory cell infiltration, compared to minimal or no expansion and inflammation in the NS group. The intimal layer shows no change in size. The star represents the lumen. All measurements are in micrometers.

We believe this outcome was expected, as aneurysm formation typically undergoes a period of acute inflammation before progressing to chronic inflammation and developing into an organized ILT. With the addition of ANG II, the aneurysm becomes less stable, and ruptures sooner compared to the PE+Pa model with only BAPN, which demonstrated more ILT formation (**Supplementary Fig.4**). In the future, by capturing the ruptured AAA tissue, we aim to better characterize ILT formation in ruptured AAAs. We have revised the manuscript to include these explanations and provided a more detailed description of the day 6 histology (**Page 12, lines 19-22**).

c. The discussion (page 9, lines 24-28) suggests that PPE + PAP enhances rupture, but based on the data, rupture appears to depend primarily on the addition of BAPN and BAPN + ANG II (17% and 93% rupture rates, respectively). Please revise this statement to more accurately reflect the findings.

We have revised this statement as follows: "Lastly, the ANG II pump addition, achieved a remarkable 93% rate of infrarenal AAA ruptures in the PE+Pa combination, which was significantly higher than when using PE or Pa alone." (**page 13, lines 10-12**). Additionally, please refer to our previous response regarding the nature of the PE+Pa combination and its impact on AAA severity, regardless of the addition of BAPN and ANG II.

d. Given the very rapid rupture kinetics of the NvRM, please discuss whether and how this model truly reflects human AAA rupture pathophysiology and chronic AAA progression, as stated in the conclusion.

Thank you for your insightful comment; it raises a critical topic. Our objective in developing this model is to gain insights into the kinetics and pathophysiology of aneurysm rupture in the infrarenal aorta. We aim to study detection methods and identify potential preventive measures for the acute event of aneurysm rupture.

We have now included data showing that, on day 6, our model exhibits de novo AAAs (**Fig. 4B**). These established aneurysms subsequently rupture, creating a unique rupture rate model ideal for studying this severe complication. Understanding AAA growth and progression is a separate but complementary aspect. For studies specifically focused on preventing growth and progression, our model utilizing BAPN alone (**Fig. 2**) may better recapitulate the chronic nature of the disease. While it also results in rupture, it does so at a lower rate. We have acknowledged the rapid rupture kinetics as a limitation of the now PE+Pa model in our discussion. We understand this might not fully represent the human

pathophysiology of AAA growth and rupture. Please refer to the discussion section in **page 15, lines 30&31 and page 16, lines 1-3**.

e. The comparison to the TGF β model is interesting, but it is not clear why the use of four inducers simultaneously (including systemic agents such as BAPN and Ang II) would reduce, rather than increase, concerns about off-target effects. Please clarify this point.

Thank you for your insightful comment. We now clarify our discussion regarding the "off-target effects" of TGF- β model. TGF- β is a critical regulator with numerous downstream effects and significantly impacts AAA pathophysiology. Therefore, using TGF- β to induce AAAs might serve as a confounding factor when evaluating treatments, potentially augmenting or diminishing the effects of the drug being tested. Moreover, TGF- β mutations are found only in a specific subset of AAA patients and do not represent the broader AAA population. We have included these important points in the revised discussion section and adjusted the "off-target effects" statement accordingly. (**Page 14, Lines 28-31 and Page 15, Lines 1-2**). Additionally, our model employs widely researched factors such as BAPN and ANG II, which have contributed to numerous findings and have been supported with literature for many years.^{1,2,6-13}

4. For several models, the justification of concentrations used is unclear due to either missing references or lack of explanation. This information is critical for reproducibility and for other researchers to apply these models:

a. Was the PPE used at a 100% concentration? i.e. 50 μ L of 10.3 mg/mL? Please indicate in the methods. Additionally, page 12 line 18 in the methods says "As previously reported," to describe the PPE model, yet no study is referenced. The Bhamidipati et al. study referenced in line 14 did not use the cotton ball method and only 10 μ L elastase for 10 min incubation.

Thank you for pointing out the correction. We have now clarified that we used 100% concentration in all our models, which we now clarify in our methods section. Additionally, we provided a more detailed explanation of how the 5'PE model was reproduced (**Page 6, lines 22-26**). While the concept of 100% concentration was initially introduced by Bhamidipati et al.,¹³ we did not use this model as our control because it is not a rupture model. Instead, we used the 5-minute 5 μ L model by Lu et al.,⁴ which included BAPN and demonstrated rupture events. However, in our hands, this did not result in any ruptures. We have now clarified these points in the methods section (**Page 6, lines 22-26**) and revised all references accordingly.

b. Similarly, line 28 on the same page again says "As previously described," with no reference, please reference appropriately.

We have revised and now included a reference for this (**Page 6, line 18**).

c. In the methods under “Chronic model of AAA in male C57BL/6 mice”, “0.3% BAPN in water” was used in this study, referencing a publication of an AAA rat model (ref 15), when the study that established this model used 0.2% BAPN (ref 31). Please clarify the reasoning for the increase in concentration in text or reference the publication with the protocol followed.

Thank you for your comment. In our previous publications utilizing an intraluminal elastase model, we consistently used 0.3% BAPN in both rat and mouse experiments.^{3,11,14,15} Since our primary goal for this study was to increase the likelihood of rupture events, we employed the highest BAPN concentration possible in the drinking water, in contrast to the 0.2% used in other studies.⁴ We now provide **references 15 and 22** in our main manuscript (**Page 6, line 9**), which represent the first-ever use of 0.3% BAPN in a rat AAA model and our recent publication using this same concentration in a mouse AAA model, respectively. We have consistently demonstrated that this concentration is effective, feasible, and does not lead to major complications.

d. The angiotensin II was indicated to be used at 2000 ng/kg/min, but both studies referenced used ApoE KO mice. In fact, 1000 ng/kg/min is the typical in the ApoE KO model as established by Daugherty et al. (ref 25). Please explain the reasoning behind the concentration used or reference the publication with the protocol the authors followed for C57BL/6 mice.

We have now included the correct reference for the 2000 ng/kg/min dosage used by Fashandi et al. (**Reference 17**), a dose that we have also utilized and published in our own studies in mice (**Reference 22**), as previously mentioned.³ **Please refer to the updated references in Page 6, line 18.** Again, our aim for this study was to maximize the rupture rate, so we used the highest concentration that has already been tested, published, and proven to be safe and consistent.^{3,5} These references have now been included in the methods section detailing AAA rupture induction.

Minor Comments:

1. Figure 1 D, in the PPE model VVG: there is hardly any elastin degradation shows, please choose a more representative image that matches the quantification in Fig 1E.

Thanks for the suggestion. We have revised this. Please refer to updated **Fig. 1D**.

a. Also, in the same figure, some times ns is indicated while other times there is nothing indicated, which should also mean ns. Please stay consistent across panels to avoid confusion.

Thanks for the comment. We have revised accordingly in all figures.

2. In the NvRM, it is unclear what type of ruptures occurred, were they all abdominal or were there any thoracic ones observed? Please indicate with AA ruptures for clarity. Also, in line 5, page 7, "NvRM" is used without clear indication of the acronym. Please write out "Novel rupture model (NvRM)" at first mention.

All ruptures were in the infrarenal abdominal segment; we did not find any thoracic pathology (**Fig. 3D&E**). We have revised NvRM to PE+Pa. Additionally, we now clarified that these were AA ruptures in the **Fig. 2 and 3 legends (Page 24, lines 6-7 and Page 25, lines 8-9)**.

3. The elastase-induced model is referred to as PPE throughout most of the results, but in Fig. 4 and the related text, "E+B" appears. Please keep terminology consistent (e.g., PPE+BAPN) as elsewhere.

We have updated the group name from E+B to 5'PE, based on previous suggestions. Please refer to the explanations and responses provided above for further details.

4. In the results, page 8, line 6, it says "elastin degradation was significantly decreased in both E+B and NvRM groups when compared to controls, but qualitative analysis demonstrated a more severe degradation in the NvRM group". Please double check for accuracy, since the figures appear to show more severe degradation, not "decreased".

We have revised this to elastin degradation was significantly **increased**, this was a typo (**Page 12, line 23**).

5. In Supplementary figure 3B, the average weight of mice in PAP group appears to drop from week 3 to week 4 then recovers in week 5. Please comment on this. Additionally, please indicate the n numbers that comprise the data in Supp Fig 3, as well as Supp Fig 2 B-D, and Supp Fig 4.

We have revised our supplementary figures and methodology section, under Statistics and Reproducibility to include all the suggestions (**Page 8, lines 29-31**): "The sample size (n) for each experiment is provided in corresponding figure legends. See Supplementary Data for all source data, raw numerical results, and original zymography gels for MMP2 and MMP9." While there is no clear explanation for the observed drop in weight, we agree it is interesting. Fortunately, these mice subsequently recovered their weight and remained active and healthy for the duration of the study.

6. In Supplementary figure 4, the legend doesn't specify which model the images are from, please indicate.

We now clarify that this figure comes from the Pa group that received BAPN only. This has been revised and now included in the **Supplementary Fig. 4** legend.

7. In the methods section, it is indicated that mice used in the study were 6-8 weeks old. Typically, wildtype mice used in elastase-induced AAA studies, including ones cited here, are 8-12 weeks old at AAA induction. Please comment to this.

This is an excellent point. The age range of 6-8 week-old mice has been used in previous studies, such as the one by Fashandi et al., which utilized 7-week-old mice.² This age range is advantageous for experiments requiring extended observation periods, such as our 6-week BAPN experiment (**Fig. 2**). It is important to note that in younger mice (3 weeks old), BAPN and ANG II can be more effective but may also promote off-target aortic degeneration.¹⁶ Therefore, using mice older than 6 weeks is preferable, and we were able to demonstrate the absence of thoracic aneurysms or dissections (**Fig. 3E and Supplementary Fig. 3**).

8. In the methods, under "Chronic model of AAA in male C57BL/6 mice", it is indicated that "In addition to either Pap or PPE+Pap exposures, two groups of mice also underwent β -aminopropionitrile (BAPN) administration through drinking water (0.3% BAPN in water) to promote AAA rupture", however the publication also presents PPE + BAPN, as well as controls. Please update the information to be comprehensive of all the groups.

This information has been revised and updated in the methods section (**Page 6, lines 6-12**), as well as in **Table 1** and **Supplementary Table 2**.

9. In Table 1, under the "Rupture AAA" model, the volume for elastase in "BAPN+E" is listed as "5," which appears to be a typo, likely 50 μ L. Please verify.

We apologize for the confusion. This is not a typo. The previous E+B or BAPN+E model, now revised to "5'PE," was reproduced using 5 microliters for 5 minutes without the use of a cotton ball, as detailed in the methods section (**Page 6, lines 22-26**) and based on a previous publication.⁴

10. In Figure 6, The use of "C" in parentheses (e.g., "C BAPN," "C Pap") is unclear. If "C" stands for "Chronic," the placement before the inducer is confusing since BAPN is what results in the chronic aspect. Please clarify in the legend or remove.

Thank you for the suggestion. We now clarified the confusing references for the groups and have adopted a more consistent nomenclature. Please refer to the explanation provided above.

Overall, this is an interesting and potentially valuable study. Addressing the points above

will strengthen the manuscript and make it more accessible and reproducible for others in the field.

Reviewer #2 (Remarks to the Author):

The authors propose a novel mouse model of abdominal aortic aneurysm (AAA) that combines established induction methods to create either a chronically enlarged aneurysm model or an early rupture model. One of the major reasons why basic research on AAA treatment has not translated successfully into clinical practice is that pharmacological agents effective in mouse models often fail to show efficacy in humans. In particular, there are few reliable models for aneurysm rupture, and the model proposed by the authors could greatly advance our understanding of AAA pathophysiology.

1. In the chronic and rupture models, there appears to be no significant difference in the levels of the cytokines and proteolytic enzymes presented, suggesting that other factors may contribute to rupture. Please discuss this point further, including potential differences between mice and humans. If blood pressure is involved, please include blood pressure data as well.

Thank you for your insightful comment. We would like to clarify that, unfortunately, we did not test cytokines and proteolytic enzymes in the chronic model. Additionally, a direct comparison between the day 14 model (**Fig. 1**) and the rupture model harvested at day 6 (**Fig. 4**) is not feasible due to the different timelines. This means we cannot infer that other factors contribute to rupture based on cytokine levels from these models. We agree that multiple factors are likely to contribute to AAA rupture.

Regarding blood pressure, we did not record it ourselves and we have state this in the discussion section (**Page 16, lines 7-10**). However, it is well-established and highly consistent that ANG II pumps promote the development of abdominal aortic aneurysms.^{7,10} Importantly, not all studies that use ANG II pumps rely on blood pressure monitoring because the majority of the literature does not support the notion that blood pressure changes, if any, are a major contributor to AAA development in the mouse model.⁶ Additionally, we have explained above that the effects of ANG II are dose-dependent in terms of rupture promotion, as previously demonstrated by others.²

In the chronic model, the aneurysm enlarged substantially, and the presence of intraluminal thrombus resembles findings observed in human AAAs. Some mouse models show spontaneous regression of aneurysms over time, whereas in humans, aneurysms always

continue to expand and eventually rupture. Does this model show aneurysm regression or rupture after more than six weeks of follow-up?

Thank you for your comment; this is indeed an important aspect to consider. In our study, we did not observe any indication of aneurysm regression at any of our examined timelines. We were quite surprised to find that the aneurysms continued to grow even at the 6 week mark. We did not extend our follow-up beyond 6 weeks since our study's aim was to develop a reproducible and feasible rupture model that allows investigators to measure molecular and therapeutic targets efficiently.

There have been studies, such as those utilizing BAPN, which demonstrated aneurysm progression up to 100 days post-surgery.⁴ While we believe that extending the observation period beyond 6 weeks might be impractical for sustained experiments or treatment models, our current data suggests that the aneurysms would likely continue to grow until rupture. We agree that this is an area worthy of further study, but it was beyond the scope of our current study.

Figure 1C: The aneurysm appears to form acutely within 20 minutes after chemical exposure. This differs from human AAAs, which develop over a chronic course. How do the authors interpret this difference?

Thank you for your thoughtful comment. We interpreted the acute dilation of the aortic diameter as a response to the chemical incubation. This dilation is more pronounced with higher doses and longer incubation times, as well as the specific nature of the chemical incubation, as clearly shown in Fig. 4B. We do not consider this an acutely formed aneurysm, but rather an initial dilation due to chemical injury, which likely promotes subsequent growth and AAA establishment. We have revised the terminology to "dilation" after 20 minutes instead of "increase in aneurysm diameter." It is important to highlight that a higher initial dilation, when exposed using the same technique and time with different chemicals, does not correlate with rupture events, as shown in our results section (**Page 11, lines 26-29**): "Furthermore, the PE+Pa combination leads to a significant increase in rupture events, with an absolute risk increase of 73% and 33% compared to Pa and PE alone ($p = 0.009$ and $p = 0.03$, respectively), despite exhibiting similar post-exposure aortic diameters (Figure 3C)."

Page 4, line 26: The manuscript states that "aortic diameter continued to grow over a 14-day follow-up," but only the diameters at the time of chemical exposure and at day 14 are presented. It is therefore unclear whether the aneurysm enlarged rapidly or gradually. Please provide intermediate measurements to clarify the growth pattern.

Thank you for your great comment. We now include an intermediate measurement of the aneurysm at day 6 to clarify the growth pattern (**Fig. 4B**). This confirms AAA growth at an intermediate time point before the final assessment at day 14.

How many mice were used in each experiment? Across all experiments, were there any mice that died during or after surgery from causes other than rupture?

Figure 2B: The number of mice appears to vary among groups. Is this due to deaths during the study? Were the aortic diameters of mice that died from rupture included in the analysis?

Thank you for the great questions. We used a total of 125 mice, as summarized in **Fig. 5**. We have also detailed the number of mice used in each group for each cohort in the figure captions, as expressed in the revised methods section (**Page 9, lines 29-31**) to enhance reproducibility. Additionally, the exact values and raw data are included in the supplementary data to ensure reproducibility.

In the rupture model where ANG II was further included (**Fig. 3 and 4**), we observed some post-operative complications, defined as any fatal complication within the first 5 days of surgical AAA induction (please refer to the figure below). No post-operative complications were observed in the other cohorts. Moreover, the aortic diameters of mice that died from rupture were included in the analysis. We now clarify in our figure legends that some rupture tissues were not assessed for histology or protein due to these complications.

Post-operative complications from day 0 to day 5 following AAA surgical induction. Most complications occurred on day 3 post-operatively, particularly in the PE and Pa groups.

Figures 3F, 3G, Supplementary Figures 2B and 2D: How were the numbers of inflammatory cells and vascular smooth muscle cells quantified? Please describe in detail in the Methods section which staining method was used, which region of the tissue was analyzed, and under what microscopic field the counts were performed.

We have now included a more detailed explanation of the histopathological analysis (**Page 7, lines 25-30 and page 8, lines 1-2**), which was previously performed and validated by a clinical pathologist in our lab.¹⁵

Figures 3 and 4: Data on aortic diameter at the time of rupture are missing. In human AAAs, rupture is rare when the diameter is less than twice normal, and the risk increases with further enlargement. Since rupture in this model may occur before significant aortic dilation, measurement of aortic diameter at rupture is essential.

Thank you for your comment. Please refer to the explanations above, where we show de novo AAA rupture based on diameter measurements taken on day 6, prior to rupture. This data is now included in the main **Fig. 4B**.

Figures 1M and 4F: The y-axis values for IL-17A differ markedly between the two graphs.

Thanks for the suggestion and great attention to detail. This is now revised. Please refer to updated **Fig. 1M and 4K**.

Page 9, line 24: It cannot be concluded that inflammatory markers peak at day 6 based only on the data from that time point. Time-course data are needed to support this statement.

Thank you for your comment. We agree that we cannot conclusively state this. We revised the discussion section to reflect this and have changed this statement (**Page 14, Lines 17-18**).

References

1. Angiotensin II promotes atherosclerotic lesions and aneurysms in apolipoprotein E-deficient mice.
2. Fashandi, A. Z. *et al.* A novel reproducible model of aortic aneurysm rupture. *Surgery (United States)* 163, 397–403 (2018).
3. Sastriques-Dunlop, S. *et al.* Ketosis prevents abdominal aortic aneurysm rupture through C–C chemokine receptor type 2 downregulation and enhanced extracellular matrix balance. *Sci Rep* 14, 1438 (2024).
4. Lu, G. *et al.* A novel chronic advanced stage abdominal aortic aneurysm murine model. *J Vasc Surg* 66, 232-242.e4 (2017).
5. Fashandi, A. Z. A Novel Reproducible Model of Aortic Aneurysm Rupture. *Physiol Behav* 176, 139–148 (2017).
6. Sawada, H., Lu, H. S., Cassis, L. A. & Daugherty, A. Twenty Years of Studying AngII (Angiotensin II)-Induced Abdominal Aortic Pathologies in Mice: Continuing Questions and Challenges to Provide Insight into the Human Disease. *Arteriosclerosis, Thrombosis, and Vascular Biology* vol. 42 277–288 Preprint at <https://doi.org/10.1161/ATVBAHA.121.317058> (2022).
7. Lu, H. *et al.* Subcutaneous angiotensin II infusion using osmotic pumps induces aortic aneurysms in mice. *Journal of Visualized Experiments* 2015, (2015).
8. Fashandi, A. Z. *et al.* A novel reproducible model of aortic aneurysm rupture. *Surgery (United States)* 163, 397–403 (2018).

9. Lysgaard Poulsen, J., Stubbe, J. & Lindholt, J. S. Animal Models Used to Explore Abdominal Aortic Aneurysms: A Systematic Review. *European Journal of Vascular and Endovascular Surgery* vol. 52 487–499 Preprint at <https://doi.org/10.1016/j.ejvs.2016.07.004> (2016).
10. Alan Daugherty, Michael W. Manning, and L. A. C. ANG II promotes atherosclerotic lesions and aneurysms in apolipoprotein E-deficient mice. *Journal of Clinical Investigation* 105, (2000).
11. English, S. J. *et al.* Increased 18f-fdg uptake is predictive of rupture in a novel rat abdominal aortic aneurysm rupture model. *Ann Surg* 261, 395–404 (2015).
12. Liu, Z., Koga, J.-I. & Wang, B. *Progress in Murine Models of Ruptured Abdominal Aortic Aneurysm.*
13. Bhamidipati, C. M. *et al.* Development of a novel murine model of aortic aneurysms using peri-adventitial elastase. in *Surgery (United States)* vol. 152 238–246 (2012).
14. English, S. J. *et al.* CCR2 Positron Emission Tomography for the Assessment of Abdominal Aortic Aneurysm Inflammation and Rupture Prediction. *Circ Cardiovasc Imaging* 13, E009889 (2020).
15. Elizondo-Benedetto, S. *et al.* Chemokine Receptor 2 Is a Theranostic Biomarker for Abdominal Aortic Aneurysms. *JACC Basic Transl Sci* <https://doi.org/10.1016/j.jacbts.2025.02.010> (2025) doi:10.1016/j.jacbts.2025.02.010.
16. Sawada, H., Beckner, Z. A., Ito, S., Daugherty, A. & Lu, H. S. β -Aminopropionitrile-induced aortic aneurysm and dissection in mice. *JVS-Vascular Science* vol. 3 64–72 Preprint at <https://doi.org/10.1016/j.jvssci.2021.12.002> (2022).